# Regulatory T cells trigger effector T cell DNA damage and senescence caused by metabolic competition

Xia Liu[1], Wei Mo[1], Jian Ye[1], Lingyun Li[1], Yanping Zhang[2], Eddy C. Hsueh[2], Daniel F. Hoft[1] & Guangyong Peng [1]

Defining the suppressive mechanisms used by regulatory T (Treg) cells is critical for the development of effective strategies for treating tumors and chronic infections. The molecular processes that occur in responder T cells that are suppressed by Treg cells are unclear. Here we show that human Treg cells initiate DNA damage in effector T cells caused by metabolic competition during cross-talk, resulting in senescence and functional changes that are molecularly distinct from anergy and exhaustion. ERK1/2 and p38 signaling cooperate with STAT1 and STAT3 to control Treg-induced effector T-cell senescence. Human Treg-induced T-cell senescence can be prevented via inhibition of the DNA damage response and/or STAT signaling in T-cell adoptive transfer mouse models. These studies identify molecular mechanisms of human Treg cell suppression and indicate that targeting Treg-induced T-cell senescence is a checkpoint for immunotherapy against cancer and other diseases associated with Treg cells.

[1] Department of Internal Medicine, Division of Infectious Diseases, Allergy & Immunology, Saint Louis University School of Medicine, St Louis, MO 63104, USA. [2] Department of Surgery, Division of General Surgery, Saint Louis University School of Medicine, St Louis, MO 63110, USA. Xia Liu and Wei Mo contributed equally to this work. Correspondence and requests for materials should be addressed to G.P. (email: guangyong.peng@health.slu.edu)

Treg cells have a central function in the prevention of autoimmunity and maintenance of immune homeostasis[1,2]. However, Treg cells also have deleterious effects by aiding the persistence of infectious pathogens and blocking effective anti-tumor immunity[3]. It is established that the Treg-mediated tumor suppressive microenvironment presents a major barrier for successful immunotherapy[4]. Defining the suppressive mechanisms used by different types of tumor-infiltrating Treg cells is essential for the development of strategies to treat human cancers. Depletion of Treg cells and/or prevention of Treg cell suppressive activities through immune checkpoint blockade of CTLA-4 or PD1/PDL1, have been utilized in animal models and clinical trials, and have yielded promising results[5–7]. However, these strategies can concurrently eliminate activated effector T cells and their activities, and the success rates are still limited and varied[8–10]. Therefore, alternative strategies targeting more specific checkpoint molecules or interrupting tolerogenic pathways for Treg cell suppression are urgently needed.

Although progress has been made in understanding the molecules and mechanisms that Treg cells use to mediate suppression, the precise suppressive mechanisms induced by human Treg cells are unclear[11,12]. In previous studies, we discovered that both human CD4+CD25hiFoxP3+ naturally occurring Treg (nTreg) and tumor-derived γδ Treg cells induce responder T-cell senescence as a suppressive mechanism[13,14]. Senescent T cells induced by Treg cells have phenotypic changes, including expression of senescence-associated-β-galactosidase (SA-β-Gal)[13,14], downregulation of co-stimulatory molecules CD27 and CD28[13,15,16], and promotion of cell cycle and growth arrest in G0/G1 phase. Importantly, both senescent CD4+ and CD8+ T cells induced by Treg cells have potent suppressive activities[13,14]. In addition, we identified that human tumor cells also use induction of T-cell senescence as a suppressive mechanism in tumor microenvironments[17,18]. These studies clearly suggest that senescent T cells are critical mediators and amplifiers of immune suppression mediated by Treg cells, and that blockage of Treg-induced senescence in responder immune cells is a critical checkpoint to control Treg cell suppression. Defining the cellular and molecular processes of generation and functional alterations of senescent T cells induced by human Treg cells should result in the development of new strategies for control of Treg-mediated suppression and restoration of effector T-cell function.

In addition to senescence, there are other two states of T-cell dysfunction, exhaustion, and anergy, which have been identified in chronic infection, cancer, and autoimmune diseases[19,20]. Both exhausted and anergic T cells have defective effector functions, but with distinct regulatory mechanisms. T-cell exhaustion was initially described in chronic virus infections with increased expression of a panel of inhibitory receptors, including PD-1, LAG-3, CD244 (2B4), CD160, CD57, KLRG1, and Tim-3[21,22]. Other studies suggest that exhausted T cells also exist in patients with various types of cancer[23–25]. Anergic T cells are induced by antigenic stimulation, but without sufficient co-stimulation and/or high co-inhibition, causing hyporesponsiveness and low IL-2 production[26,27]. Given that Treg-induced senescent T cells are phenotypically and functionally similar to anergic and exhausted T cells, whether senescent T cells are also anergic or exhausted is unclear. Furthermore, whether senescent T cells are an independent and unique T-cell lineage is unknown. A better understanding of the relationships and differences between senescent T cells, and exhausted or anergic T cells will not only clarify the definition of these cells, but also should provide alternative strategies and targets for clinical immunotherapy.

In our current study, we explore the molecular mechanisms of human Treg cell suppressive function. Treg cells initiate the nuclear kinase ataxia-telangiectasia mutated protein (ATM)-associated DNA damage response in responder T cells triggered by glucose competition during cross-talk, resulting in responder T-cell senescence and functional changes. Furthermore, we identify that MAPK ERK1/2 and p38 signaling functionally cooperate with transcription factors STAT1/STAT3 to control responder T-cell senescence induced by human Treg cells. In addition, our studies indicate that senescent T cells are not functionally exhausted and are different from anergic T cells. Importantly, we perform in vivo studies in mice and show that Treg-induced T cell senescence can be prevented by inhibiting the DNA damage response and/or STAT signaling. These studies identify molecular processes responsible for human Treg suppression, and provide proof-of-concept for therapeutic reversal of Treg-induced immune suppression.

## Results

**Human Treg cells initiate DNA damage in responder T cells.** We have recently discovered that human nTreg and tumor-derived γδ Treg cells induce responder naive/effector CD4+ T-cell senescence with altered phenotypic and functions[13,14]. However, the molecular mechanisms involved in the differential induction of senescence and dysfunction of responder T cells are still unknown. There are several mechanisms potentially involved in the induction of cellular senescence. The induction of DNA damage is the key molecular process in senescent cells, and the nuclear kinase ATM is the chief inducer of the DNA damage response[28,29]. We therefore investigated whether induction of DNA damage is the main trigger for human Treg-induced senescence in treated T cells. We first determined protein expression and phosphorylation of ATM in T cells. We found that CD4+ T cells co-cultured with or without control CD4+CD25− T cells had no or minor expression of phosphorylated ATM. In contrast, nTreg treatment significantly induced phosphorylation of ATM in responder CD4+ T cells (Fig. 1a). We also detected activation of the other key DNA damage response proteins, including ATM substrates H2AX and 53BP1, as well as the downstream target checkpoint kinase 2 (CHK2). As expected, significant activation and phosphorylation of CHK2, H2AX, and 53BP1 was induced in responder CD4+ T cells treated with nTreg cells but not with control CD4+CD25− T cells (Fig. 1a). In addition, nTreg treatment significantly induced activation and phosphorylation of those DNA damage molecules in naive CD8+ T cells (Fig. 1b). We further confirmed that the increased phosphorylation levels of the DNA damage molecules were not due to increases of their total protein levels in senescent T cells induced by nTreg cells (Supplementary Fig. 1). Besides the nTreg cells, we extended our findings to breast tumor-derived γδ Treg cells[14,30–32]. We found that co-culture with γδ Treg cells also markedly induced activation and phosphorylation of ATM and DNA damage molecules CHK2, H2AX, and 53BP1 in both responder CD4+ and CD8+ T cells (Fig. 1c, d). However, CD4+CD25− T cells and γδ2 T cells purified from peripheral blood of healthy donors did not promote the activation of those molecules in responder T cells. To further confirm that induction of ATM-associated DNA damage is the cause for human Treg-induced responder T-cell senescence, we determined whether we can prevent the senescence induction in T cells through functional blockage of ATM-induced DNA damage in responder T cells using the loss-of-function approach with the ATM-specific pharmacological inhibitor KU55933[13,30,33]. We found that treatment with KU55933 in responder T cells can significantly prevent the activation and phosphorylation of ATM, CHK2, H2AX, and 53BP1 in responder CD4+ and CD8+ T cells treated with nTreg cells (Supplementary Fig. 2a). Importantly, pretreatment with KU55933 in responder T cells dramatically prevented

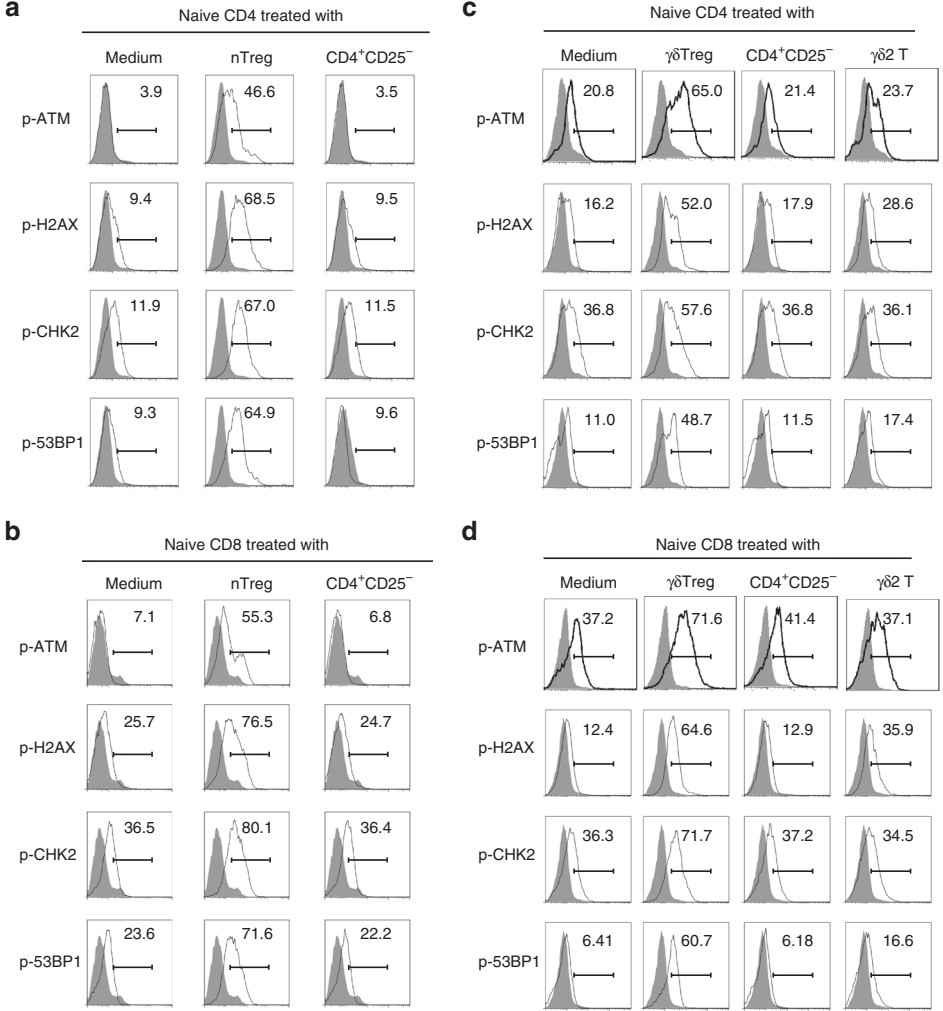

**Fig. 1** Human nTreg and γδ Treg cells induce DNA damage response in responder T cells. **a**, **b** Phosphorylated activation of ATM and other associated molecules H2AX, 53BP1, and CHK2 in both naive CD4⁺ T (in **a**) and naive CD8⁺ T (in **b**) cells treated with CD4⁺CD25ʰⁱFoxP3⁺ Treg (nTreg) cells. Anti-CD3 activated naive T cells were treated with nTreg or CD4⁺CD25⁻ effector T cells (control) for 3 days. The p-ATM, p-H2AX, p-53BP1, and p-CHK2 expression in treated naive CD4⁺ and CD8⁺ T cells were analyzed by the flow cytometry. **c**, **d** Upregulated phosphorylated ATM and other associated molecules H2AX, 53BP1, and CHK2 were determined in pre-activated naive CD4⁺ (in **c**) and CD8⁺ (in **d**) T cells treated with γδ Treg cells for 3 days by the flow cytometry. Anti-CD3-activated naive T cells treated with CD4⁺CD25⁻ and γδ2 effector T cells were served as controls

the induction of SA-β-Gal⁺ T-cell populations in both CD4⁺ and CD8⁺ T cells induced by Treg cells (Fig. 2a, b). Furthermore, ATM-signaling blockage partially restored the loss of co-stimulatory molecules, CD27, and CD28, in senescent CD4⁺ and CD8⁺ T cells induced by Treg cells (Fig. 2c, d). In addition, we determined the functional role of ATM-associated DNA damage in the development of T-cell senescence induced by tumor-derived γδ Treg cells. Consistent with the results shown in nTreg cells (Fig. 2a, b), functional blockage of ATM signaling with the ATM-specific inhibitor KU55933 significantly reduced the induction of senescent T cells in both CD4⁺ and CD8⁺ T cells mediated by γδ Treg cells (Supplementary Fig. 2b). In addition to ATM, the kinase ATM- and Rad3-related (ATR) is important to orchestrate the DNA damage response in senescent cells[28,29]. Notably, co-culture with nTreg but not control CD4⁺CD25⁻ T cells also promoted the phosphorylation of ATR in responder T cells, suggesting that ATR may involve the regulation of Treg-induced T-cell senescence (Supplementary Fig. 3a). Collectively, these results clearly indicate that initiation of DNA damage is critical and the main cause for the induction of T-cell senescence and dysfunction mediated by human Treg cells.

Besides phenotypic changes, Treg cell treatment significantly increased cell cycle regulatory molecules p53, p21, and p16 expression in responder T cells[13,14]. Given that senescent growth arrest is established and maintained by p53/p21 and/or p16/pRB tumor suppressor pathways[34], we thus investigated the interactions between DNA damage-associated proteins (phosphorylated-ATM, -H2AX, -53BP1, and -CHK2) and cell cycle regulatory molecules in senescent T cells induced by Treg cells using the immunofluorescence in situ analyses. We observed substantially increased expression of phosphorylated-ATM, phosphorylated-H2AX, phosphorylated-53BP1, and phosphorylated-CHK2 in responder T cells treated with Treg cells, but not with control T cells (Fig. 2e, f). Furthermore, phosphorylated-ATM, -H2AX, -53BP1, and -CHK2 molecules were co-expressed with the increased p53, p21, or p16 in Treg-induced senescent T cells (Fig. 2e, f), suggesting the causative relationships between DNA damage response and cell cycle regulation during T-cell senescence mediated by human Treg cells.

**Glucose competition triggers T-cell senescence and DNA damage**. We have previously shown that tumor-derived

endogenous cAMP is responsible for the induction of T-cell senescence mediated by tumor cells[17,18]. cAMP is also crucial for nTreg cell-mediated suppression[35]. We therefore investigated the role of cAMP in Treg-induced senescence and DNA damage, using the functional blockage assays with inhibitors of 7-ddA (an inhibitor specific for adenylate cyclase) and H89 (an inhibitor specific for PKA). Surprisingly, treatment of nTreg cells with 7-ddA and H89 did not decrease the percentages of senescent T-cell

population in responder CD4[+] T cells co-cultured with nTreg cells (Supplementary Fig. 3b). These results suggest that human tumor cells and Treg cells may utilize different mediators/molecules to induce responder T-cell senescence, and that cAMP is only a critical mediator responsible for tumor-induced T-cell senescence.

Studies suggest that tumor cells and tumor-infiltrating T cells (TIL) compete for glucose within the tumor suppressive

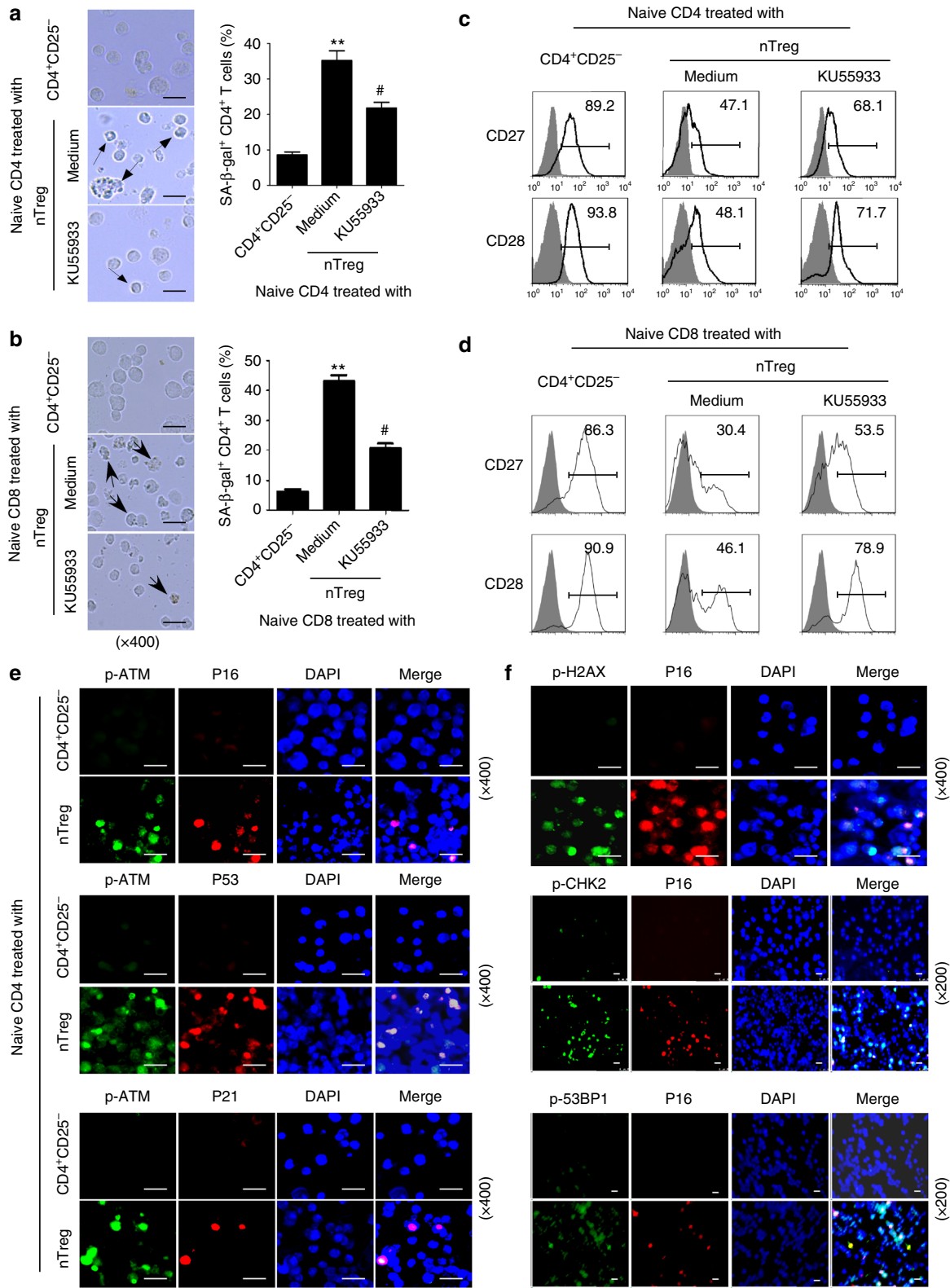

microenvironment, which is a driver for tumor suppression and progression[36,37]. We therefore hypothesized that the lack of enough glucose highly consumed by Treg cells may result in responder T-cell senescence and DNA damage. Given that movement of glucose in and out of cells is controlled by facilitative glucose transporters (Gluts), we first compared expression levels of Glut1 and Glut3, two dominant Gluts in CD4+CD25− effector and Treg cells[38,39]. We found that both nTreg and γδ Treg cells have much higher expression levels of Glut1 and Glut3, compared with effector T cells (Fig. 3a). To further test the possibility that Treg cells have heightened glucose consumption, we determined the glucose uptake ability of Treg, naive, and effector T cells, using a fluorescent glucose analog 2-NBDG labeling assay[39,40]. Our results clearly showed that nTreg and tumor-derived γδ Treg cells have a significantly higher glucose uptake than control naive CD4+ and CD8+ T cells, as well as CD4+CD25− effector T cells (Fig. 3b). Furthermore, we confirmed that nTreg highly consumed glucose in the medium during 3 days of culture compared with naive CD4+ T cells, indicating a heightened metabolic activity of Treg cells. In addition, co-culture with nTreg cells but not control CD4+CD25− effector T cells with naive CD4+ T cells dramatically decreased the glucose level in the co-culture medium (Fig. 3c). These results collectively suggest that human Treg cells have heightened glucose consumption distinct from naive and effector T cells. We then investigated whether low concentrations of glucose can directly induce T-cell senescence. Anti-CD3-activated naive CD4+ T cells were cultured with selected concentrations of glucose (0–11 mM) for 5 days and then studied for cell senescence. As expected, we found that the increased senescent CD4+ T-cell populations were markedly induced in the medium with low concentrations of glucose in a dose-dependent manner (Supplementary Fig. 4a). Furthermore, low concentrations of glucose in the medium also significantly induced downregulation of CD27 and CD28, promotion of cell cycle regulatory molecule expression, and phosphorylation of ATM, H2AX, CHK2, and 53BP1 in CD4+ T cells (Supplementary Fig. 4b, c, d). In addition, pretreatment with KU55933 in responder T cells dramatically prevented the induction of SA-β-Gal+ T-cell populations in CD4+ T cells induced by low glucose, suggesting that induction of ATM-associated DNA damage is the cause for low glucose-induced T-cell senescence (Supplementary Fig. 4e). AMP-activated protein kinase (AMPK), an important nutrient and energy sensor, is activated by metabolic stresses[41]. Recent studies suggest that AMPK is also activated by DNA damage agents directly regulating T-cell senescence[42,43]. To determine the causative link between glucose shortage and T-cell senescence, we observed significantly increased phosphorylation of AMPK in responder T cells treated with Treg cells, but not with control T cells (Fig. 3d). Furthermore, phosphorylated-AMPK was also promoted in T cells with low concentrations of glucose (Fig. 3e). These results strongly suggest that Treg-induced T-cell senescence is involved in AMPK-associated glucose metabolic regulation.

To identify whether glucose metabolic processes are important for regulatory function of human Treg cells, we determined the Treg suppressive capacities on responder T-cell proliferation and senescence induction in the presence of pharmaceutical inhibitors targeting glucose transport and glycolysis[13,14]. We found that blockages of both glucose transporter and glycolysis with respective inhibitors phloretin and 2-deoxy-D-glucose (2-DG), can significantly reverse Treg suppressive effect on the proliferation of responder T cells and markedly decrease responder T-cell senescence induced by both nTreg and breast tumor-derived γδ cells (Fig. 3f, g). To further determine whether the competition of glucose consumption between Treg and responder T cells is the main reason for Treg-induced T-cell senescence and DNA damage, we then investigated whether addition of glucose can prevent Treg-induced senescence in responder T cells during their interactions. As shown in Fig. 3h, i, addition of high concentrations of glucose (25 and 35 mM) dramatically prevented the cell senescence and downregulated the phosphorylated activation of ATM, H2AX, 53BP1, and CHK2 in responder T cells mediated by nTreg cells. Collectively, these results clearly indicate that glucose competition between Treg and responder effector T cells triggers cell senescence and DNA damage in responder T cells during their interactions.

**STAT1 and SATA3 control Treg-induced T-cell senescence.** We identified that human CD4+CD25hiFoxP3+ Treg cells induce selective modulation of specific MAPK p38 and ERK1/2 signaling pathways in responder T cells that control the molecular process of Treg-induced T-cell senescence[13]. To further explore the molecular process responsible for Treg-induced T-cell senescence, we performed the transcriptional analysis of Treg-induced senescent T cells (accession code GSE38765). Enrichment tests of gene ontology (GO) categories and associated $p$ values were calculated for the genes most significantly altered in Treg-treated T cells during the process of senescence induction using online software (http://portal.genego.com). Our results suggested that c-Myc, SP1, STAT1, and STAT3 are the most significantly altered candidate transcription factors involved in the regulation of T-cell senescence (Fig. 4a). To further validate the transcriptional analysis results, we determined the expression levels and phosphorylated activation patterns of the four transcription factors in human Treg-induced senescent T cells. We observed that nTreg cells significantly enhanced activation and phosphorylation of STAT1 and STAT3 in treated naive CD4+ T cells (Fig. 4b). However, nTreg treatment did not induce marked alterations of transcription factors c-Myc and SP1. These results suggested that transcription factors STAT1 and STAT3 might be involved in the regulation of responder T-cell senescence induced by human Treg cells.

**Fig. 2** Blocking ATM-associated DNA damage response prevents human Treg-induced T-cell senescence. **a**, **b** Pretreatment of naive CD4+ (in a) and CD8+ (in **b**) T cells with ATM inhibitor KU55933 significantly decreased senescent T-cell populations in responder T cells induced by nTreg cells. Anti-CD3 activated T cells were pretreated with or without KU55933 (10 μM) for 1 day, and then co-cultured with human nTreg cells for 3 days. SA-β-Gal expression in naive CD4+ and CD8+ T cells was determined with SA-β-Gal staining. The SA-β-Gal positive T cells were identified with dark blue granules as indicated by the arrows. Scale bar: 20 μm. Data shown in the right panels are mean ± SD from three independent experiments, and paired $t$ test was performed. **p < 0.01, compared with the control CD4+CD25− and KU55933 treatment groups. #p < 0.01, compared with the control CD4+CD25− T cell treatment group. **c**, **d** Pretreatment of responder T cells with ATM inhibitor KU55933 markedly restored CD27 and CD28 expression in Treg-treated naive CD4+ (in **c**) and CD8+ (in **d**) T cells. CD27 and CD28 expression in treated naive CD4+ T cells as described in **a** and **b** were analyzed by the flow cytometry. **e**, **f** DNA damage response molecules and cell cycle regulatory molecules are co-expressed in human Treg-induced senescent CD4+ T cells as described in Fig. 1. Immunofluoresence double staining with antibodies against p-ATM, p-H2AX, p-53BP1, or p-CHK2 with anti-P16, P53, or P21antibodies in the same slide from responder T cells treated with Treg or control CD4+CD25− effector T cells. Scale bar: 20 μm

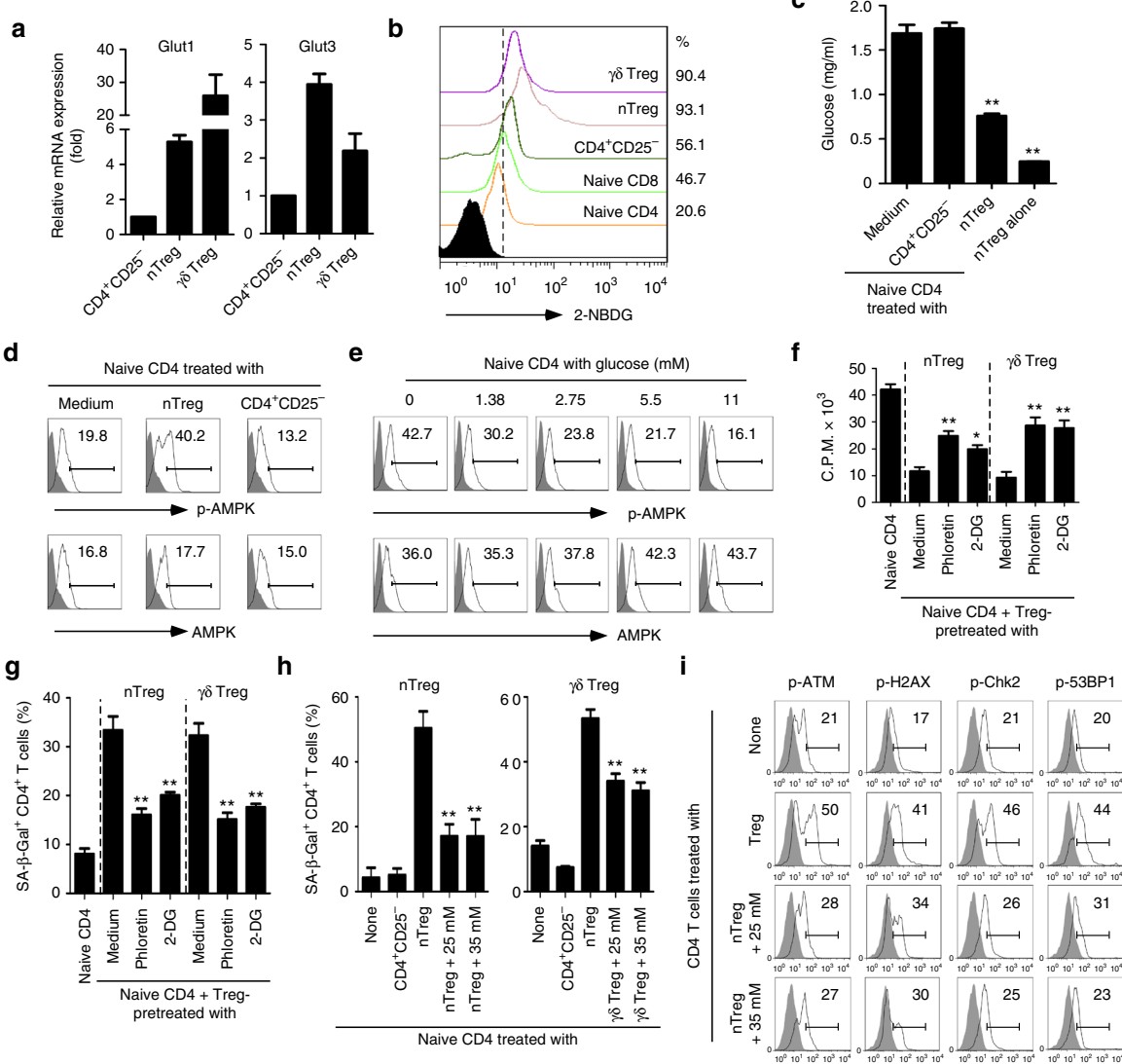

**Fig. 3** Treg-mediated glucose competition causes responder T-cell senescence and DNA damage. **a** Higher expression levels of glucose transporters (Glut1 and Glut3) in Treg cells than that of effector T cells analyzed by real-time PCR. Expression levels of each gene were normalized to β-actin expression and adjusted to the levels in CD4+CD25− cells only (served as 1). Data shown are mean ± SD from four independent donors. **b** Human nTreg and tumor-derived γδ Treg cells have a high glucose uptake capacity determined by the flow cytometry after addition of 2-NBDG. Results shown are a representative of four independent experiments. **c** Co-culture of nTreg with naïve CD4+ T cells for 3 days significantly decreased glucose levels in the culture medium. **$p < 0.01$, compared with naive T cells in medium only group by unpaired $t$ test. **d** Increased phosphorylation of AMPK was induced in naive CD4+ T cells treated with nTreg cells. The p-AMPK and total AMPK expression in treated T cells for 3 days was analyzed by the flow cytometry. **e** Low concentrations of glucose significantly promoted phosphorylation of AMPK in anti-CD3-activated naive CD4+ T cells for 3 days using the flow cytometry analysis. **f, g** Inhibition of glucose transport and glycolysis dramatically blocked Treg suppressive capacity on T-cell proliferation (in **f**) and responder T-cell senescence induction (in **g**). Treg cells were pretreated with inhibitors phloretin (2 μM) or 2-DG (1 mM) for 1 day, and then co-cultured with naive CD4+ T cells for 3 days. Data shown are mean ± SD from three independent experiments.*$p < 0.05$ and **$p < 0.01$, compared with the medium only group by paired $t$ test. **h** Addition of high concentrations of glucose markedly rescued responder T-cell senescence induced by Treg cells determined with SA-β-Gal staining after 3-day co-culture. Data shown are mean ± SD from three independent experiments. **$p < 0.01$, compared with the Treg-treated group with normal concentration of glucose (11 mM) using paired $t$ test. **i** High concentrations of glucose significantly prevented the DNA damage-associated molecule activation in responder T cells induced by nTreg cells for 3 days using the flow cytometry analyses

Recent studies have shown that STAT1 and STAT3 are also involved in cellular senescence induced by $H_2O_2$, stress, and radiation[44,45]. Therefore, we determined whether we can prevent the senescence induction in T cells through functional blockage of STAT1 and STAT3 signaling in responder T cells. Anti-CD3-activated naive CD4+ T cells were pretreated with or without JAK (upstream in the STAT signaling pathway) and/or STAT specific pharmacological inhibitors, including Tyrphostin AG490 (30 μM, JAK inhibitor), MTA (5 μM, STAT1 inhibitor), or S3I-201 (10

μM, STAT3 inhibitor). The STAT activity-blocked or -unblocked T cells were then cultured with human Treg cells for different time points (0, 3, 5 days) and studied for STAT activation and cellular senescence induction. As expected, pretreatment of responder naive CD4+ T cells with AG490 almost completely blocked the Treg-induced STAT1 and STAT3 activation in CD4+ T cells (Fig. 4c). Furthermore, pretreatment of T cells with MTA or S3I-201 also dramatically suppressed the phosphorylation of STAT1 and STAT3 in Treg-treated naive CD4+ T cells,

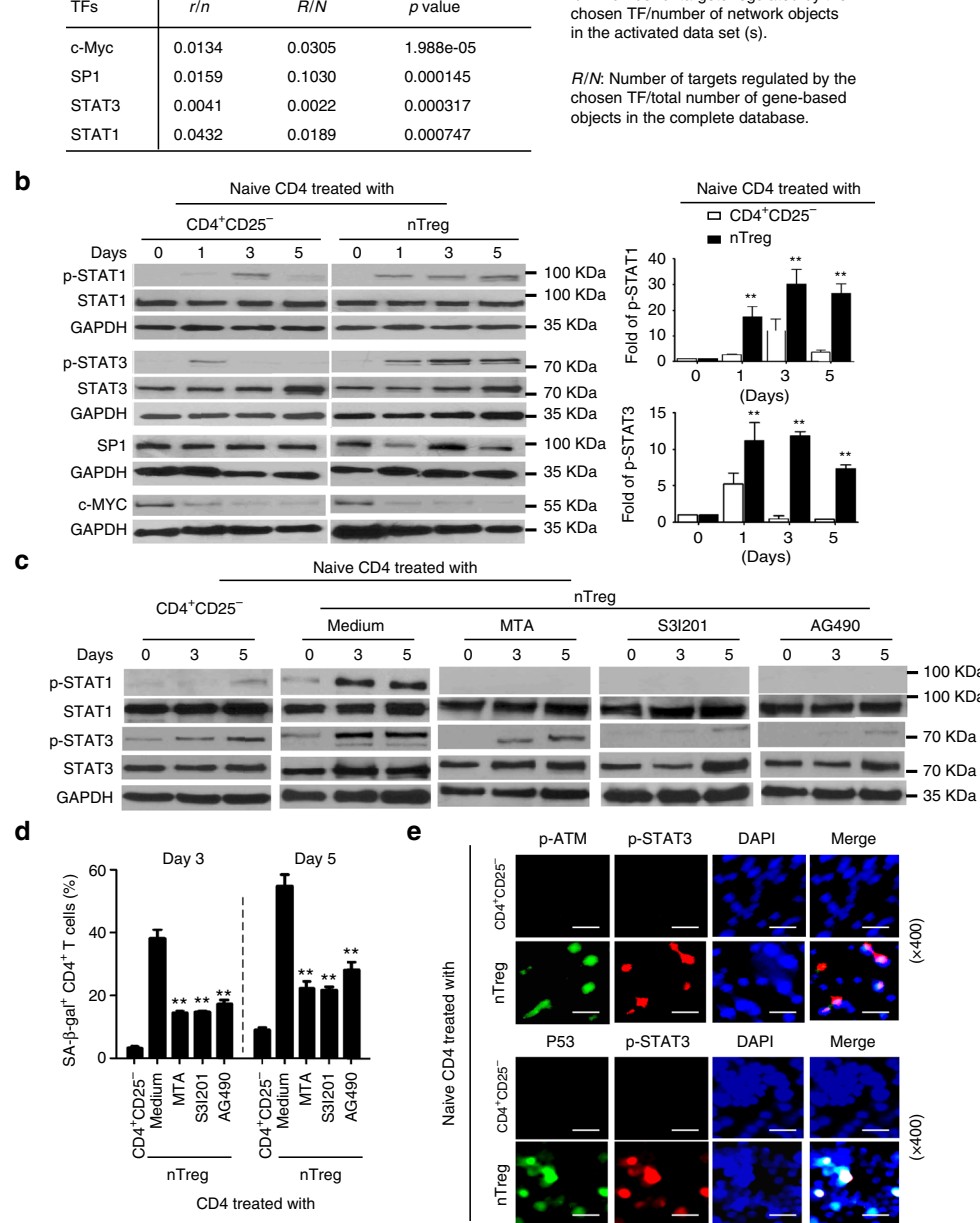

**Fig. 4** STAT1 and SATA3 control responder T-cell senescence mediated by human Treg cells. **a** Transcriptome analyses showed that c-Myc, SP1, STAT1, and STAT3 were the most significantly altered transcription factors in Treg-induced senescent CD8[+] T cells. Gene alterations involved in transcription factors and related signaling pathways were identified and ranked in naive CD8[+] T cells after treatment with or without nTreg cells at different time points. **b** Phosphorylated activation of STAT1 and STAT3 in senescent CD4[+] T cells induced by nTreg cells. Results from western blot analyses of phosphorylated activation of STAT1, STAT3, SP1, and c-Myc are shown in the left panel. Phosphorylated STAT1 and STAT3 protein levels shown in the right histogram were quantitatively analyzed and compared against the GAPDH expression levels with a densitometer. Results shown in the histogram are mean ± SD from three independent experiments. $^{**}p < 0.01$ compared with the CD4[+]CD25[-] control group using paired $t$ test. **c**, **d** Inhibition of STAT1 or STAT3 signaling by specific pharmacological inhibitors significantly suppressed STAT1 and STAT3 phosphorylation and decreased SA-β-Gal expression in T cells treated by Treg cells. Anti-CD3-activated naïve CD4[+] T cells were pretreated with or without JAK and/or STAT inhibitors tyrphostin AG490 (30 μM, JAK inhibitor), MTA (5 μM, STAT1 inhibitor) or S3I-201 (10 μM, STAT3 inhibitor) for 1 day. The STAT activity-blocked or -unblocked T cells were then cultured with human Treg cells at a ratio of 4:1 for different time points (0, 3, 5 days) and studied for STAT1 and STAT3 activation with western blot analysis (in **c**) and SA-β-Gal expression (in **d**). Data shown are mean ± SD from three independent experiments, and paired $t$ test was performed. $^{**}p < 0.01$, compared with the group not treated with inhibitors. **e** Phosphorylated STAT3 was co-expressed with phosphorylated ATM and P53 in senescent T cells induced by human Treg cells. Immunofluoresence double staining with antibodies against STAT3, and p-ATM or P53 in the same slide of responder T cells treated with Treg or control CD4[+]CD25[-] effector T cells. Scale bar: 20 μm

respectively. Notably, STAT3 signaling inhibitor S3I-201 has potent inhibition on both STAT1 and STAT3 phosphorylation, suggesting that STAT3 signaling may regulate STAT1 activation during the T-cell senescence process (Fig. 4c). In addition, inhibitors AG490, MTA and S3I-201 significantly reduced Treg-induced senescence in naïve CD4$^+$ T cells (Fig. 4d). We also investigated the interaction and co-expression of STAT signaling with DNA damage and cell cycle regulatory molecules in Treg-induced senescent T cells. We found that phosphorylated STAT3 was significantly induced and co-localized with the phosphorylated-ATM and p53 in Treg-induced senescent T cells (Fig. 4e). These results indicate that STAT1 and STAT3-mediated signaling are also critical and important in regulating cell senescence in Treg-treated responder T cells.

**MAPK cooperates with STAT to regulate T-cell senescence.** Our studies have shown that both MAPK and STAT1/3 signaling pathways are involved in the regulation of Treg-induced T-cell senescence (Fig. 4)[13]. Therefore, we next investigated how these two signaling pathways interact and work together during the process of ATM-associated T-cell senescence. We found that Treg treatment significantly induced expression and phosphorylation of ERK1/2 and p38 signaling in responder T cells at different time points of co-culture. However, functional blockage of ATM activation with the ATM-specific inhibitor KU55933 markedly prevented the phosphorylation of ERK1/2 and p38 in senescent T cells mediated by human Treg cells (Fig. 5a). In addition, phosphorylated-ERK and p38 molecules were co-expressed with the p53, or p16 in Treg-induced senescent T cells (Fig. 5b), suggesting the causative relationships between ATM-associated DNA damage response, MAPK signaling activation, and cell cycle regulation during T-cell senescence.

We next explored the reciprocal interactions and relationships between MAPK signaling and STAT1/3 signaling in T-cell senescence. We utilized the loss-of-function strategy with specific inhibitors U0126 and SB203580, and then determined whether blockage of the MAPK signaling can directly affect STAT signaling activation induced by Treg cells. We observed that treatment with U0126 or SB203580 almost abolished activation and phosphorylation of STAT1 and STAT3 during the process of T-cell senescence mediated by human Treg cells (Fig. 5c). In addition, blockage of ATM signaling with KU55933 significantly decreased STAT1 and STAT3 phosphorylation and activation in Treg-induced senescent T cells (Fig. 5d). To further explore their reciprocal interaction during T-cell senescence, we performed the immunofluorescence analysis to visualize the interaction and incorporation between MAPK signaling (phosphorated-p38 or -ERK) versus STAT signaling (phosphorated-STAT3), in Treg-induced senescent T cells. Consistent with the above results, we found significantly increased phosphorylated activation of ERK, p38 and STAT3 in responder T cells treated with Treg cells, but not with control T cells (Fig. 5e). In addition, phosphorylated-p38 and -ERK molecules were co-expressed with the phosphorylated-STAT3 in Treg-induced senescent T cells (Fig. 5e). These studies clearly indicate the importance and causative regulations between ATM-associated DNA damage initiation, MAPK and STAT signaling activation during the T-cell senescence progression induced by human Treg cells.

**Senescent T cells induced by Treg cells are not exhausted.** Exhausted T cells have recently been identified in certain types of patients, including in chronic infection, cancer, and autoimmune diseases[19,20]. Exhausted T cells are characterized as the increased expression of a panel of inhibitory receptors, including PD-1, LAG-3, CD244 (2B4), CD160, CD57, KLRG1, and Tim-3[21,22].

Furthermore, exhausted T cells are functionally deficient with decreased effctor cytokine expression, such as IL-2, IFN-γ, and TNF, and defective effector functions[21,22]. To exclude the possibility that senescent T cells induced by human Treg cells are also exhausted T cells, we determined T-cell exhaustion markers on senescent T cells induced by human Treg cells. We found that both naive CD4$^+$ and CD8$^+$ T cells treated with nTreg cells dramatically decreased CD28 expression, compared with T cells treated with CD4$^+$CD25$^-$ effector T cells (Fig. 6a and Supplementary Fig. 5)[13,14]. Furthermore, nTreg treatment induced mild expression of PD1, CD160, and CD57 in treated CD4$^+$ T cells, but not in CD8$^+$ T cells. However, nTreg cell treatment did not significantly promote the expression of 2B4, KLRG1, and Tim-3 in both senescent CD4$^+$ and CD8$^+$ T cells during the senescence development, consistent with the hypothesis that these senescent T cells are not exhausted T cells (Fig. 6a and Supplementary Fig. 5). We also extended these studies to breast tumor-derived γδ Treg cells. We observed very similar results as shown in nTreg cells, that γδ Treg cells induced significant loss of CD28 expression and moderate expression of CD57 and KLRG1 in responder CD4$^+$ and CD8$^+$ T cells, but did not induce increased expression of PD1, CD160, and Tim-3 in both senescent CD4$^+$ and CD8$^+$ T cells (Supplementary Fig. 6). Given that CD57 and KLRG1 have been reported as markers for senescent T cells[46,47], our results indicate that Treg-induced senescent T cells do not exhibit exhausted T-cell phenotypic inhibitory receptors.

We also investigated whether senescent T cells induced by human Treg cells are functionally exhausted with decreased effecter cytokine production. We found that treatment with nTreg cells significantly promoted the mRNA expression of a panel of pre-inflammatory and effector cytokines at different time points, including IL-1β, IL-2, IL-6, IL-8, TNF, and IFN-γ (Fig. 6b). In addition, increased levels of IL-10 mRNA expression were also induced in responder CD4$^+$ T cells mediated by nTreg cells during the senescence induction (Fig. 6b). Those cytokine mRNA expression results have also been confirmed at the protein level in senescent T cells induced by both nTreg and γδ Treg cells in our previous studies[13,14]. Our results indicate that senescent T cells induced by human Treg cells are still functionally active rather than exhausted T cells, secreting pre-inflammatory and effector cytokines and performing suppressive function[13,14,18].

**Treg-induced senescent T cells differ from anergic T cells.** Treg-induced senescent T cells possess well-described functional phenotypes, such as nonproliferation in response to antigenic stimulation, similar to those in anergic T cells. We next asked whether Treg-treated responder T cells have similar gene expression profiles associated with anergic T cells. We compared the transcriptional profiling difference in anergy-associated genes between anergic T cells and Treg-induced senescent T cells. We selected 28 genes reported to be associated with anergy for the comparison between these gene expression in human Treg-induced senescent T cells[48,49]. These gene expression changes were kinetically assessed at different time points, including early (4–8 h), middle (24–48 h), and late (72 h), during the conversion of naive CD8$^+$ T cell into senescent T cells. As shown in Fig. 7a, the genes were divided into three groups based on the gene expression patterns compared with those in anergic T cells[48,49]. Genes listed in group 1 were significantly downregulated in Treg-induced senescent T cells, but reported to be upregulated in anergic T cells. Genes listed in group 2 were significantly upregulated in Treg-induced senescent T cells, but reported to be downregulated in anergic T cells. Among the 28 genes, only 10 genes had similar expression alterations comparing Treg-induced senescent T cells and anergic T cells (Group 3), but the others

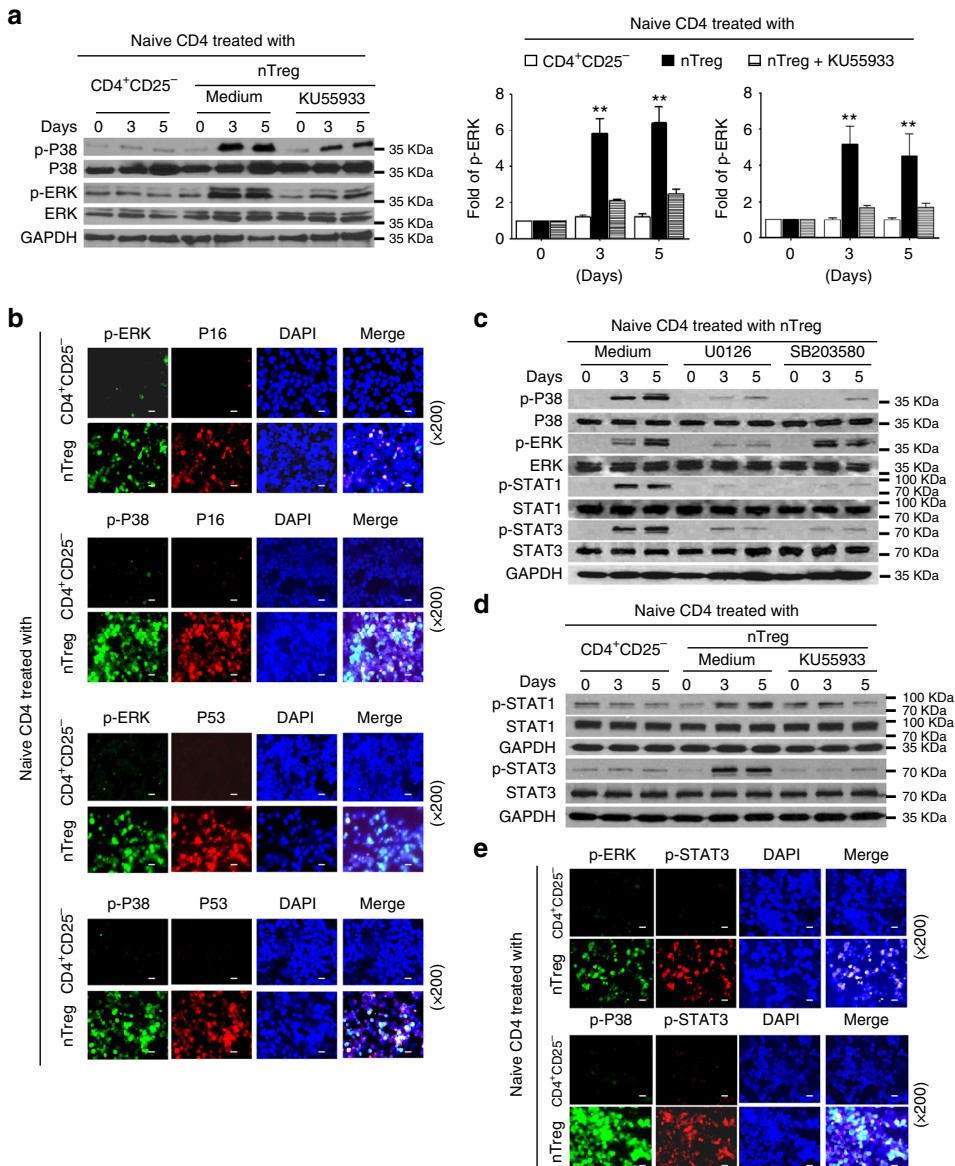

**Fig. 5** MAPK cooperates with STAT to control T-cell senescence mediated by Treg cells. **a** Pretreatment of naive CD4$^+$ T cells with an ATM inhibitor KU55933 significantly prevented P38 and ERK phosphorylation in responder T cells treated by nTreg cells using western blot analyses. Anti-CD3 activated CD4$^+$ T cells were pretreated with or without KU55933 (10 μM) for 1 day, and then co-cultured with nTreg cells for different time points. Protein levels of phosphorylated P38 and ERK shown in the right histogram were quantitatively analyzed and compared against the GAPDH expression levels with a densitometer. Results shown in the histogram are mean ± SD from three independent experiments. **$p < 0.01$, compared with the medium only and KU55933 treatment groups by paired $t$ test. **b** Phosphorylated P38 and ERK were co-expressed with cell cycle regulatory molecules P16 and P53 in human Treg-induced senescent CD4$^+$ T cells. Immunofluoresence double staining with antibodies against p-ERK and p-P38 with anti-P16 or P53 antibodies in the same slide of responder T cells treated with Treg or control CD4$^+$CD25$^-$ effector T cells. Scale bar: 20 μm. **c** Inhibition of ERK1/2 or p38 signaling pathways by specific pharmacological inhibitors markedly blocked the STAT1 and STAT3 phosphorylation in responder T cells treated by Treg cells. Anti-CD3 activated CD4$^+$ T cells were pretreated with inhibitors U0126 or SB203580 (10 μM) for 1 day, and then co-cultured with nTreg cells for different time points. Phosphorylation of p38, ERK, STAT1 and STAT3 was determined by western blot analyses. **d** Pretreatment of naive CD4$^+$ T cells with an ATM inhibitor KU55933 significantly prevented STAT1 and STAT3 phosphorylation in responder T cells treated by Treg cells. Cell treatment and procedure is identical as described in **a**. Phosphorylation of STAT1 and STAT3 was determined by western blot analyses. **e** Phosphorylated P38 and ERK were co-expressed with phosphorylated STAT3 in senescent CD4$^+$ T cells induced by human Treg cells. Immunofluoresence double staining with antibodies against p-ERK and p-P38 with anti-p-STAT3 in the same slide of responder T cells treated with Treg or control CD4$^+$CD25$^-$ effector T cells. Scale bar: 20 μm

were discordant comparing expression levels between the senescent and anergic T cells (Groups 1 and 2). For example, GRAIL (gene related to anergy in lymphocytes, also known RNF128) has been shown to highly express in anergic T cells, and play a critical role in T-cell anergy induction and regulation[50,51]. However, GRAIL was significantly downregulated in Treg-induced senescent T cells (Fig. 7a, Group 1). These results suggest human Treg-

induced senescent T cells have a distinct molecular signature different from that of anergic T cells. To further confirm our transcriptome microarray analysis results, we perform real-time reverse transcription-PCR (RT-PCR) to analyze 15 genes associated with T-cell anergy in human Treg-induced senescent CD4$^+$ T cells and in anergic CD4$^+$ T cells induced with the Ca$^{2+}$ ionophore ionomycin, a well-established T-cell anergy model[48].

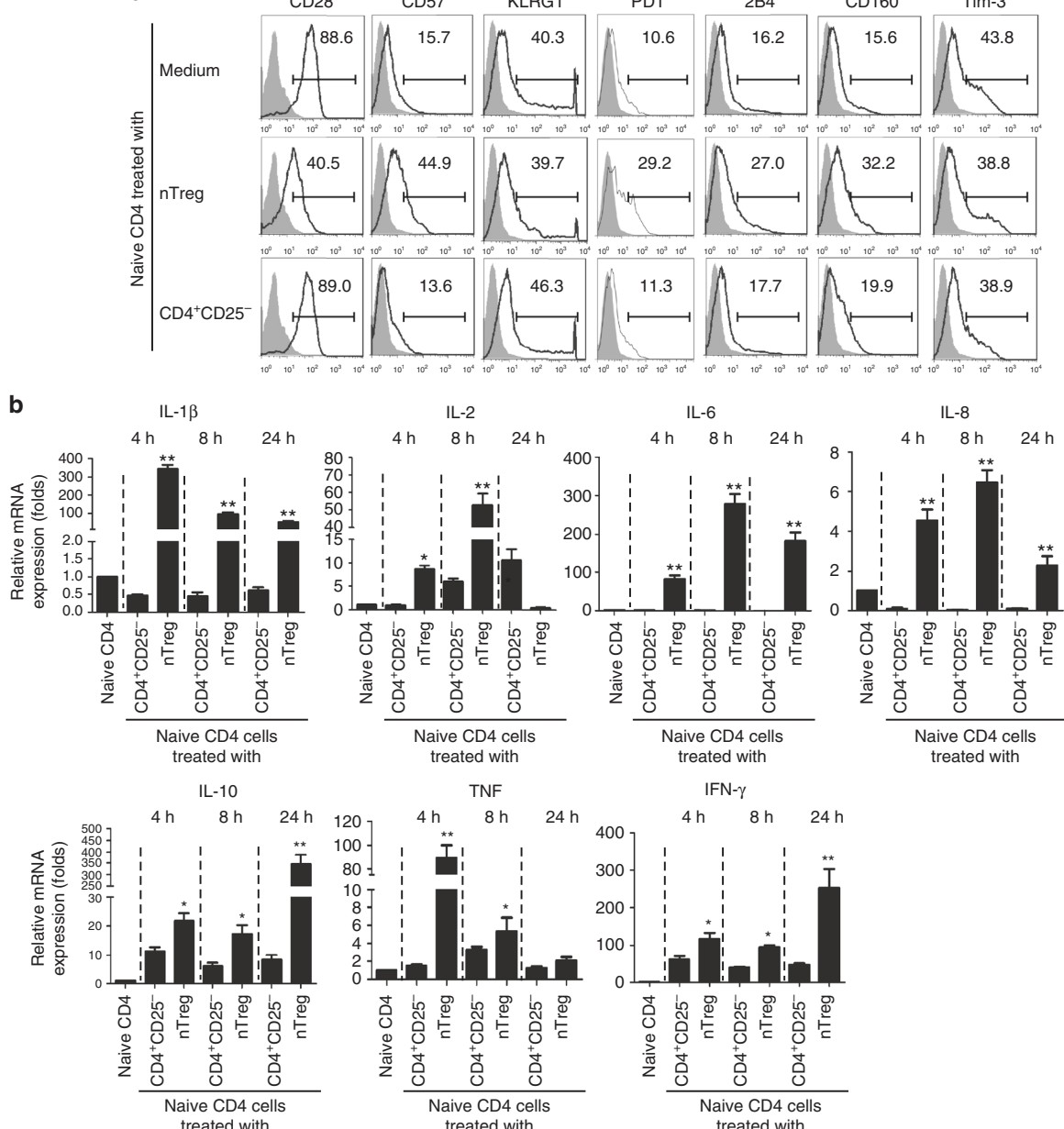

**Fig. 6** Senescent T cells induced by human Treg cells are not exhausted T cells. **a** Treg cell treatment did not promote the expression of exhaustion markers in senescent CD4+ T cells during the senescence development. Anti-CD3-pre-activated naive CD4+ T cells were co-cultured with nTreg or CD4+CD25- effector T cells (control) for 3 days, and then expression markers on treated T cells were determined by flow cytometry analyses. **b** Naive CD4+ T cells treated with CD4+CD25hiFoxP3+ Treg cells produced high levels of multiple cytokines, including IL-1β, IL-2, IL-6, IL-8, IL-10, IFN-γ, and TNF. mRNA expression levels of each cytokine were determined by the real-time PCR analyses. The expression level was normalized to GAPDH expression and adjusted to the levels in naive CD4+ T cells. Data are means ± SD from three representative naive CD4+ T cells. *$p < 0.05$ and **$p < 0.01$ compared with the CD4+CD25- treatment group determined by paired $t$ test

Importantly, the expression patterns of these 15 genes in senescent CD4+ T cells were different from that in anergic CD4+ T cells (Fig. 7b). Collectively, these results strongly suggest that Treg-induced senescent T cells are functionally active and not anergic[15].

**Prevention of Treg-induced T-cell senescence in vivo.** Given that we identified that ATM-induced DNA damage is the main cause for induction of T-cell senescence and dysfunction mediated by human Treg cells, we therefore investigated whether we can functionally block the human Treg-induced conversion of

T cells into senescent cells by inhibiting DNA damage in responder T cells in vivo in an adoptive transfer model[13,14]. We pre-activated naive CD8+ T cells and CD4+CD25hiFoxP3+ nTreg cells with anti-CD3 and anti-CD28 antibodies. Pre-activated naive CD8+ T cells were further pretreated with or without the ATM inhibitor KU55933 (10 μM) for 24 h prior to adoptive co-transfer of nTreg cells into NSG mice. The adoptively transferred CD8+ T cells were purified and recovered from different groups and organs at day 12, and analyzed for the effects of ATM inhibition on phenotypic and functional changes, as we previously described[13,14,30,33]. Consistent with our previous in vivo studies, we showed that significantly increased senescent T-cell

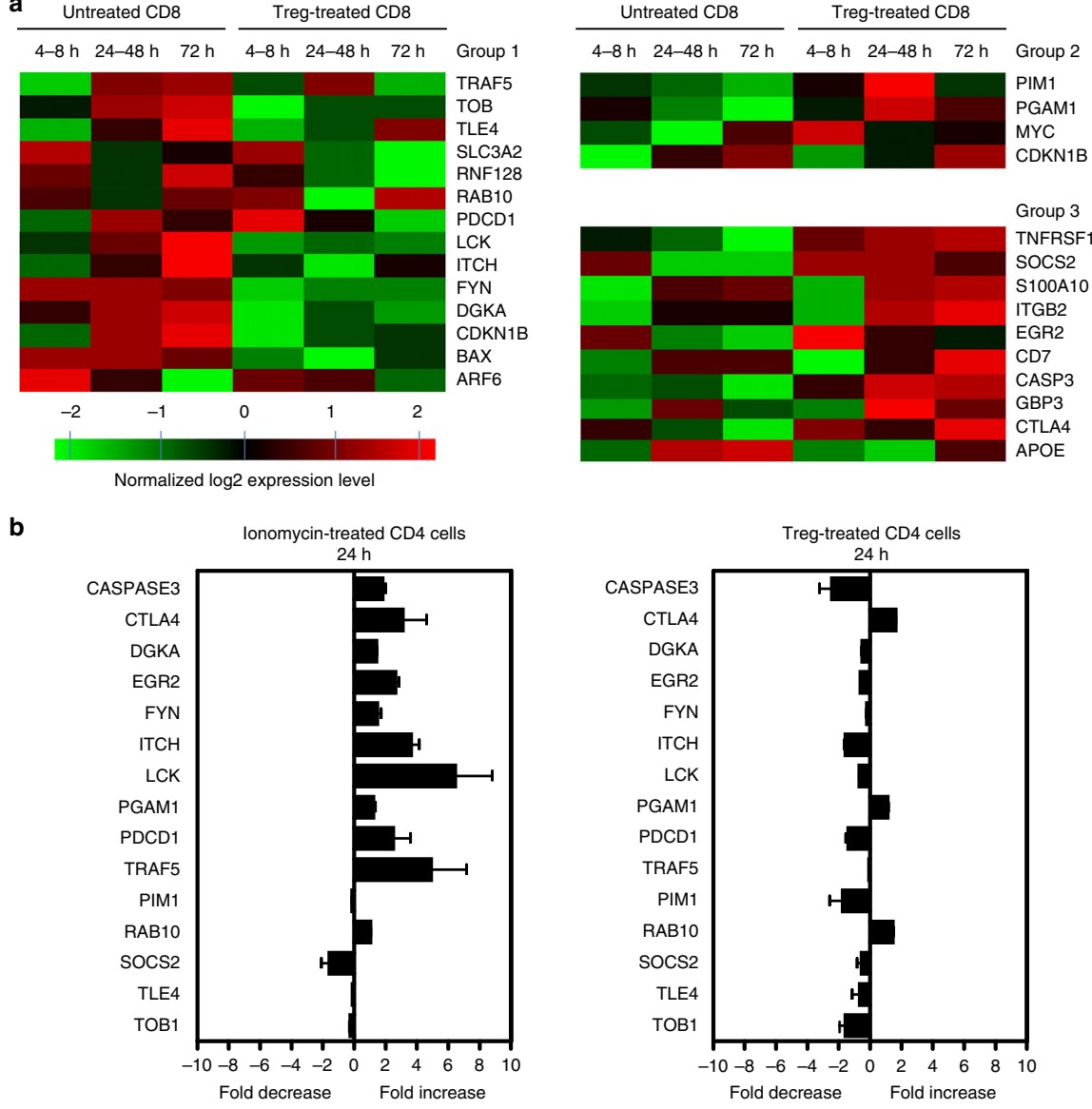

**Fig. 7** The gene signature of Treg-induced senescent T cells is different from anergic T cells. **a** Transcriptome analyses of T-cell anergy-associated genes in naive CD8$^+$ T cells after treatment with or without CD4$^+$CD25$^{hi}$FoxP3$^+$ Treg cells. Gene alterations were normalized in log2 expression levels and divided into three groups based on the gene expression patterns compared with those in anergic T cells. Genes listed in Group 1 are significantly downregulated during the process of Treg-induced senescence, but reported to be upregulated in anergic T cells. Genes listed in group 2 were significantly upregulated in Treg-induced senescent T cells, but reported to be downregulated in anergic T cells. Genes listed in Group 3 have a consistent expression trend between Treg-induced senescent T cells and anergic T cells. **b** Different expression pattern of selected anergy-associated genes in inomycin-induced anergic and Treg-induced senescent CD4$^+$ T cells. Relative mRNA expression levels of each gene in T cells were determined by real-time PCR with specific primers, and normalized to GAPDH expression. Data shown are mean ± SD from three independent experiments

populations were induced in CD8$^+$ T cells when co-transferred with CD4$^+$CD25$^{hi}$FoxP3$^+$ nTreg cells (over 40%) recovered from blood, spleen, and lymph nodes (LNs), in NSG mice (Fig. 8a). However, we observed that transferred CD8$^+$ T cells alone did not develop senescence. Furthermore, the purified CD8$^+$ T cells from spleen previously co-transferred with CD4$^+$CD25$^{hi}$FoxP3$^+$ Treg cells dramatically reduced CD28 expression compared with CD8$^+$ T cell transfer only group (Fig. 8c). In addition, the purified CD8$^+$ T cells from different organs previously co-transferred with Treg cells could suppress the proliferation of responding CD4$^+$ T cells in vitro (Fig. 8d). Importantly, pretreatment of CD8$^+$ T cells with ATM-specific inhibitor KU55933, significantly blocked senescence induction in transferred CD8$^+$ T cells showing the decreased SA-β-gal$^+$ T cells (Fig. 8a). Furthermore, blockage of

ATM signaling in CD8$^+$ T cells with KU55933 reduced CD28 surface loss and further prevented the development of suppressive activity in T cells after being co-transferred with nTreg cells in vivo (Fig. 8c, d). These results suggest that specific blockage of ATM-signaling activation in responder T cells can prevent Treg-mediated induction of T-cell senescence and subsequent immune suppression in vivo.

Our in vitro studies provided important evidence that transcription factors STAT1/3 are also critical players controlling the molecular process of responder T-cell senescence induced by human Treg cells. Thus, we next investigated whether we could prevent the induction of senescent T cells mediated by Treg cells in vivo by the specific STAT signaling inhibition in this adoptive transfer model. Activated naive CD8$^+$ T cells were pretreated with

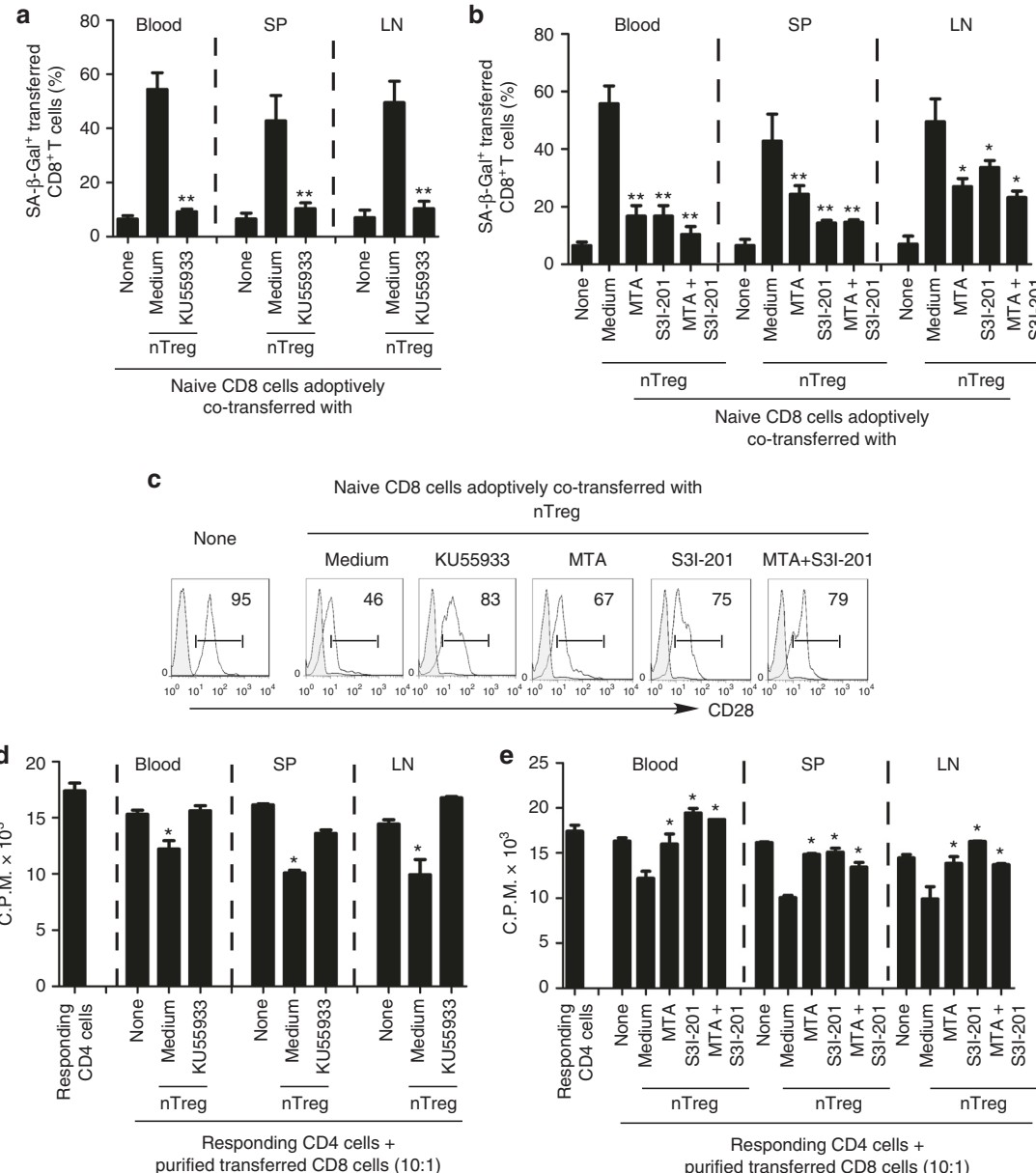

**Fig. 8** Reversal of Treg-induced T-cell senescence by blockage of ATM and STAT signaling in vivo. Pretreatment of naive T cells with ATM inhibitor or STAT signaling inhibitors before co-injection with $CD4^+CD25^{hi}FoxP3^+$ Treg cells can significantly block the induction of senescence and reverse the suppressive activity in transferred naive $CD8^+$ T cells in NSG mice. Anti-CD3 activated naive $CD8^+$ T cells ($5 \times 10^6$ per mouse) were pretreated with the ATM inhibitor KU55933, or STAT1/3 inhibitors MTA or S3I-201 for 24 h, before co-transfer with $CD4^+CD25^{hi}FoxP3^+$ Treg ($2 \times 10^6$ per mouse) into NSG mice. The transferred human $CD8^+$ T cells in different organs, including blood, lymph nodes (LN), and spleens (SP) were isolated at 12 days post injection for subsequent SA-β-Gal staining (in **a** and **b**), CD28 analysis by flow cytometry (in **c**), and suppressive activity with the $^3$H-thymidine incorporation assays (in **d** and **e**). $*p < 0.05$ and $**p < 0.01$, compared with the medium only group using the ANOVA analysis

STAT1/3 inhibitors MTA, S3I-201, or combination for 24 h before co-transfer with nTreg cells into NSG mice, following the same procedures described as above. As expected, we found pretreatment of naive $CD8^+$ T cells with these inhibitors or combination significantly blocked the induction of senescence and prevented the decrease of CD28 expression in transferred naive $CD8^+$ T cells induced by Treg cells (Fig. 8b, c). In addition, naive $CD8^+$ T cells pretreated with these inhibitors or combination did not develop suppressive activity after being co-transferred with nTreg cells (Fig. 8e). These results collectively indicate that we can functionally block the human Treg-induced conversion of responder T cells into senescent cells by inhibiting the DNA damage response and/or STAT signaling in vivo.

## Discussion

It has become clear that tumor-reactive T cells are suppressed and dysfunctional in the tumor suppressive microenvironment induced by Treg cells, representing a major obstacle for successful tumor immunotherapy[4,52]. We have recently discovered that human Treg cells can induce naive/effector T cells to become senescent T cells[13,14]. Therefore, precisely defining the molecular processes and biological changes responsible for T-cell senescence will provide novel targets for the therapeutic manipulation of the fate and function of effector T cells. In this study, we further identified that induction of DNA damage response is the cause for Treg-mediated senescence and functional changes in responder T cells, involving glucose competition and the

regulation of transcription factors STAT1/STAT3, ERK1/2, and p38 signaling. In addition, we demonstrated that senescent T cells have unique phenotypes and/or functions distinctly different from exhausted and anergic T cells. Our studies not only identify a novel mechanism responsible for human Treg-mediated immune suppression, but also provide molecular targets and checkpoint signaling to control effector T-cell fate for improved immunotherapy.

We have recently identified induction of senescence as a novel suppressive mechanism that human Treg cells use to inhibit responder naive/effector CD4+ T cells[13,14]. However, the molecular mechanisms involved in the differential induction of senescence and dysfunction of responder T cells are still unknown. Our current studies clearly demonstrated that induction of ATM-associated DNA damage is the main trigger for human Treg-induced senescence. Recent studies suggest that tumor cells and TILs compete for glucose within the tumor suppressive microenvironment, which is a driver for tumor suppression and progression[36,37]. We therefore presumed that the lack of enough glucose highly consumed by Treg cells may result in responder T-cell senescence. AMPK is an important energy sensor activated by metabolic stresses[41], which also directly regulates DNA damage and T-cell senescence[42,43]. Our data demonstrated that increased phosphorylation of AMPK in responder T cells was induced by Treg cells, indicating the molecular link between glucose shortage and T-cell senescence. In addition, using loss-of-function and gain-of-function studies, we further identified that heightened consumption of glucose mediated by Treg cells is the cause for ATM-induced DNA damage and cell senescence in responder T cells mediated by human Treg cells. Our current in vitro studies identify the causative link between Treg metabolism and the molecular process on responder T-cell fate and function mediated by glucose competition/deprivation during their cross-talk and interaction. However, whether this molecular mechanism also occurs in vivo in the physiological condition is unknown. Given that higher numbers of tumor cells than Treg cells exist in the tumor microenvironment, which also highly consume glucose, whether Treg cells still perform the similar action as in vitro co-culture on effector T cells in such pathological situation is also unknown. Our future studies will continue to explore this novel suppressive mechanism mediated by human Treg cells in vivo in different tumor models.

Senescent T-cell subset is differentiated from naive or effector T cells and has its unique phenotypes and functions[13,14,17,18]. The transcriptome analyses of Treg-induced senescent T cells have provided critical information showing that signaling pathways and certain transcription factors are the most significantly altered candidates during the process of T-cell senescence. Our previous and current studies clearly demonstrated that both MAPKs ERK1/2 and P38 signaling and transcription factors STAT1 and STAT3 are selectively involved and functionally co-operated in the process of T-cell senescence induced by human Treg cells. Studies have shown that MAPKs p38 and ERK play a major role in regulating cell cycle re-entry and oncogenic ras-induced senescence[53,54]. STAT family proteins are critical in mediating T-cell development and differentiation[44]. In addition, STAT1 and STAT3 have been demonstrated to mediate cellular senescence induced by $H_2O_2$, stress, and radiation[44,45]. Importantly, our in vitro and in vivo studies further suggest that inhibitions of specific MAPK signaling and STAT1/STAT3 pathway in responder T cells can prevent the induction of T-cell senescence mediated by Treg cells. These studies not only define the unique molecular signaling pathways that control the differentiation of senescent T cells, but also identify potential therapeutic targets

blocking Treg-mediated immune suppression and restoring effector T-cell functions for immunotherapies.

Our studies have indicated that senescent T cells are an independent T-cell lineage fate, which has its own unique phenotypic and molecular regulatory profiles. The other two states of T-cell dysfunction, exhaustion, and anergy, have been well described in chronic infection, cancer, and autoimmune diseases[19,20]. Although senescent T cells share some phenotypic and functional similarities to those in anergic and exhausted T cells, senescent T cells are not anergic and/or exhausted T cells. Besides the specific signaling pathway(s) and regulatory gene profiles, senescent T cells have a unique panel of markers, including downregulation of CD27 and CD28, high expression of SA-β-Gal, CD57 and KLRG1, but not increased expression of PD-1, 2B4, CD160, and Tim-3, which are distinct from exhausted T cells[21,22]. Furthermore, senescent T cells have a unique cytokine profile, secreting large amounts of pre-inflammatory and effector cytokines IL-1β, IL-2, IL-6, IL-8, TNF, and IFN-γ, which are also significantly different from anergic and exhausted T cells. These proinflammatory cytokines have been shown to be critical inducers of premature cell senescence[55,56]. Furthermore, IL-6 and IL-8 have also been shown to promote tumorigenesis[57,58]. Therefore, Treg-induced senescent T cells might result in induction of more senescent T cells under pathological conditions such as in the tumor suppressive microenvironments, and directly promote tumor cell growth as well. Importantly, besides bystander effects on other cells, our studies suggest that these senescent T cells induced by Treg cells possess strong suppressive activity that potently amplifies the immune suppression mediated by Treg cells[13,14,17,18]. Furthermore, Treg-induced senescent T cells secreted large amounts of suppressive cytokines IL-10 and TGF-β1, which play important roles in induction of adaptive Treg cells and maintenance of immunosuppressive microenvironments. Collectively, our studies suggest that senescent T cells induced by human Treg cells are functionally active rather than exhausted and/or anergic, and that senescent T cells might be a critical mediator and amplifier of immune suppression induced by Treg cells via both bystander and direct effects within the tumor microenvironment and in patients with chronic infections.

Besides the different types of Treg cells, our more recent studies demonstrated that human cancer cells can utilize this novel suppressive mechanism to suppress effector T cells through the induction of T-cell senescence, and maintain a suppressive tumor microenvironment[13,14,17,18]. Therefore, development of senescent T cells is a critical strategy utilized by malignant tumors for the induction of immune tolerance and suppression. Our studies strongly suggest that blockage of senescence in effector/naive T cells induced by Treg cells and tumor cells should be a critical checkpoint for enhanced anti-tumor immunity and immunotherapy. We have identified that activation of human Toll-like receptor 8 (TLR8) signaling in Treg cells and multiple types of tumor cells can prevent effector T cell senescence and reverse the suppressive activity mediated by both Treg and senescent T cells[13,14,17]. Our current in vitro and in vivo studies strongly suggest that development of senescent T cells can be reversed by blockage of ATM-associated DNA damage response or/and inhibitions of MAPK ERK1/2 and P38 signaling, as well as STAT1/STAT3 signaling, which could be alternatively useful strategies for senescence prevention[13,14]. In addition, these strategies combined with other immune checkpoint therapy, such as CTLA-4 or PD1 blockage therapy, should yield more promising results for improved cancer immunotherapy[5–7,59].

In summary, our current study provides the first evidence that human Treg cells directly suppress responder T-cell proliferation and initiate DNA damage response triggered by glucose competition, resulting in senescence and functional changes in T cells.

The ATM-associated DNA damage and cell senescence processes are molecularly controlled by MAPK ERK1/2 and p38 signaling functionally cooperating with transcription factors STAT1 and STAT3 (Supplementary Fig. 7). Senescent T cells have unique phenotypes, transcriptional profiles, gene regulation signatures, and active functions, distinct from exhausted or anergic T cells. In addition, our studies provide proof-of-concept for novel potential strategies for senescence prevention through blockade of ATM-associated DNA damage response and/or inhibition of STAT1/STAT3 signaling in effector T cells for clinical immunotherapy against cancer and chronic infections.

## Methods

**T cells and cell lines**. Buffy coats from healthy donors were obtained from the Gulf Coast Regional Blood Center at Houston. All human studies have been approved by the Saint Louis University Institutional Review Board (IRB No.15283). Peripheral blood mononuclear cells (PBMCs) were purified from buffy coats using Ficoll-Paque. Human naïve $CD4^+$ and $CD8^+$ T cells were purified from PBMCs of healthy donors by EasySep enrichment kits (StemCell Technologies). The purity of naïve T cells was >97%, as confirmed by flow cytometry. $CD4^+CD25^{hi}$ Treg cells were purified from $CD4^+$ T cells by FACS sorting after staining with anti-CD25-PE (BD Bioscience). Human γδ Treg cells were established from the primary breast cancer tissues in our laboratory and maintained in T cell medium containing 10% human AB serum and 50 μl/ml IL-2[30,31].

**Senescence-associated β-galactosidase staining**. Senescence-associated β-galactosidase (SA-β-Gal) activity in senescent T cells was detected as we previously described[13,14]. Naïve $CD4^+$ or $CD8^+$ T cells were labeled with carboxyfluorescein succinimidyl ester (CFSE) (4.5 μM), and co-cultured with Treg or control T cells at a ratio of 4:1 in anti-CD3-coated 24-well plates for 3 or 5 days. Co-cultured naïve T cells were then separated from co-cultures using fluorescence-activated cell sorting (FACS) gated on CFSE-positive populations, and then stained with SA-β-Gal staining reagent. For some experiments, the co-cultured naïve T cells were determined for SA-β-Gal expression in the presence of the following inhibitors: ATM inhibitor KU55933 (10 μM, Tocris Bioscience); STAT1 inhibitor MTA (5 μM), STAT3 inhibitor S3I201 (10 μM), and JAK2 inhibitor AG490 (30 μM) (Sigma-Aldrich); MAPK inhibitors including U0126 (10 μM), SB203580 (10 μM) and SP600125 (10 μM) (Calbiochemistry). For blockade of glucose transport and glycolysis analyses, the Treg cells were pretreated with glucose transporter and glycolysis inhibitors phloretin (2 μM) or 2-deoxy-D-glucose (2-DG, 1 mM) (Cayman Chemical) for 24 h and then co-cultured with naïve T cells in the presence of low concentrations of phloretin (0.5 μM) or 2-DG (300 μM) for 3 days.

**Western blot**. CFSE-labeled naïve $CD4^+$ T cells were co-cultured with Treg or control T cells at a ratio of 4:1 in assay medium (5% human serum, no IL-2) in the anti-CD3 antibody-precoated plates (2 μg/ml) for 0, 1, 3, or 5 days. Treated $CD4^+$ T cells were then sorted by FACS gating on CFSE-positive T cells. Whole cell lysates of the purified $CD4^+$ T cells were prepared for western blot analyses. Western blots were developed with Chemiluminescent Substrate (KPL, Maryland). The rabbit polyclonal antibodies used in western blotting are as follows: anti-ERK (#4695), anti-phospho-ERK (Thr180/Tyr182, #4511), anti-p38 (#9212), anti-phospho-p38 (Thr180/Tyr182, #4511), anti-JNK (#9252), anti-phospho-JNK (Thr183/Tyr185, #4671), anti–STAT1 (#9172), anti-phospho-STAT1 (Tyr701, #7649), anti–STAT2 (#4594), anti-phospho-STAT2 (Tyr609, #4441), anti-STAT3 (#4904), anti-phospho-STAT3 (Tyr705, #9145), anti-phospho-STAT3 (S727), anti-STAT5 (#9363), anti-phospho-STAT5 (Tyr694, #9359), anti–STAT6 (#9362), anti-phospho-STAT6 (Tyr641, #9361), anti-SP1 (#9389), anti-cMyc (#13987), and anti-GAPDH (#2118). All antibodies were purchased from Cell Signaling Technology. The dilution for primary antibody was 1:1000 for the studies. In some experiments, ATM inhibitor KU55933 (10 μM), STAT1 inhibitor MTA (5 μM), STAT3 inhibitor S3I201 (10 μM), and JAK2 inhibitor AG490 (30 μM) (Sigma-Aldrich), or MAPK inhibitors U0126 (10 μM) and SB203580 (10 μM) were present in the co-cultures and the treated T cells were then performed western blotting analyses.

**Flow cytometry**. The expression markers on T cells were determined by flow cytometry analyses after surface staining or intracellular staining with anti-human specific antibodies conjugated with PE or Alexa Flour488. These human antibodies included: anti-CD27 (clone: M-T271), anti-CD28 (clone: CD28.2), anti-2B4 (clone: 2–69), anti-PD-1 (clone: EH12.1), anti-CD57 (clone: HCD57), anti-KLRG1 (clone: 14C2A07), anti-CD160 (clone: BY55), anti-Tim-3 (clone: F38-2E2), anti-ATM (#2873), anti-H2AX (#7631), anti-CHK2 (#3440), anti-53BP (#4937), anti-AMPK (#2532), and anti-phospho-ATM (Ser1981, #5883), anti-phospho-H2AX (Ser139/Tyr142, #5438), anti-phospho-CHK2 (Thr68, #2661), anti-phospho-53BP (Ser25/29, #2674), and anti-phospho-AMPK (Thr172, #2535), which were purchased from BD Biosciences, Biolegend, or Cell Signaling Technology. The dilution of these antibodies was 1:100 for the studies. The cell gating strategy is shown in the

Supplementary Fig. 8. All stained cells were analyzed on a FACS Calibur flow cytometer (BD Bioscience) and data analyzed with FlowJo software (Tree Star).

**Indirect immunofluorescence staining**. Naïve $CD4^+$ T cells or $CD8^+$ T cells were co-cultured with Treg or control T cells labeled with CFSE (4.5 μM) at a ratio of 4:1 in anti-CD3-coated 24-well plates for 3 days. Co-cultured naïve T cells were then separated from co-cultures using FACS sorting gated on CFSE-negative populations. Treg-treated or control $CD4^+$ T cells were incubated with different combinations of rabbit anti-human antibody mixtures, including rabbit anti-human-phospho-ATM (Ser1981, #5883), anti-phospho-H2AX (Ser139/Tyr142, #5438), anti-phospho-CHK2 (Thr68, #2661), anti-phospho-53BP (Ser25/29, #2674), anti-P53 (#2527), anti-phospho-ERK (Thr180/Tyr182, #4511), anti-phospho-p38 (Thr180/Tyr182, #4511), or mouse anti-human-p16 (sc-28260, Santa Cruz), -p21 (#2947), -phospho-STAT3 (sc-815235) primary antibodies. $CD4^+$ T cells were then incubated with a mixture of two secondary antibodies Alexa Fluor488–conjugated anti-rabbit (Cell Signaling Technology) and Alexa Fluor647–conjugated anti-mouse (Biolegend), and were further counterstained with 40,6-diamidino-2-phenylindole (DAPI; Invitrogen).

**Glucose uptake**. Glucose uptake was measured following 15 min incubation of T cells with a fluorescent D-glucose analog 2-[N-(7-nitrobenz-2-oxa-1,3-diazol-4-yl) amino]-2-deoxy-D-glucose (2-NBDG) (Cayman Chemical)[40,60]. Treg and effector $CD4^+$ T cells ($1 \times 10^6$) were cultured in glucose-free T cell medium with 2% AB serum for another 30 min, and followed addition of 2-NBDG (100 μM) for 15 min. The cells were collected analyzed on a FACS Calibur flow cytometer (BD Bioscience).

**Real-time quantitative RT-PCR**. Gene expression in T cells was determined by RT-PCR after the total RNA extraction and cDNA transcription from T cells with respective kits (Invitrogen). mRNA expression level of each gene was first normalized to the gene expression of glyceraldehyde-3-phosphate dehydrogenase (GAPDH) or β-actin in the same sample, and the relative expression level was then calculated using the comparative method for quantification[61]. All gene expression levels were performed and obtained from triplicate experiments. The specific primers used for the gene expression in T cells are listed in the Supplementary Table 1.

**Transcriptome analyses of Treg-induced senescent T cells**. Anti-CD3-activated naïve $CD8^+$ T cells were co-cultured with medium only or $CD4^+CD25^{hi}FoxP3^+$ Treg cells at a 5:1 ratio for different time points, including early (4–8 h), middle (24–48 h), and late (72 h). The co-cultured naïve $CD8^+$ T cells was isolated by positive selection with microbeads (StemCell Technologies). Total RNA was purified from the Treg-treated and untreated naïve $CD8^+$ T cells using RNeasy Kit (Qiagen). Three biological replicates were generated using total RNA from the pool of five donors for each experimental condition. Transcriptome analyses of Treg-induced senescent $CD8^+$ T cells were performed using the Illumina whole-genome HumanHT-12 BeadChips[13]. To identify genes differentially expressed in naïve and Treg-induced senescent T cells, a t test was performed for each gene at the six fixed time points separately, and an F-test based on analysis of variance (ANOVA) was performed as an overall test for the combined data from the time points. p values were adjusted and $p < 0.01$ was used as a cutoff to define the significance. Hierarchical clustering was utilized to present the selected significant downregulated and upregulated genes. In addition, GO terms associated with each gene were used to characterize the functionally related genes and identify processes associated with networks of differentially expressed genes. Genes used for the comparison with anergic T cells are listed in the Supplementary Table 2. The normalized $\log_2$ expression level of each gene was calculated.

**T-cell anergy induction and anergy-associated gene analysis**. The anergic T-cell model we used is described previously[48]. In brief, $CD4^+$ T cells were treated with 1 μM ionomycin for 16 h, washed twice with RPMI 1640, and cultured in 10% FBS RPMI 1640 medium for additional 8 h. Furthermore, naïve $CD4^+$ T cells were co-cultured with CFSE-labeled $CD4^+CD25^{high}$ Treg cells at a ratio of 4:1 in the anti-CD3-coated plate for 24 and 72 h. Treg-treated $CD4^+$ T cells (CFSE negative) were sorted by FACS. Total RNAs were purified from anergic $CD4^+$ T cells and Treg-treated $CD4^+$ T cells, and the selected anergy-associated genes in these cells were determined by real-time RT-PCR analyses.

**Functional proliferation assay**. Proliferation assays were performed using a [$^3$H]-thymidine incorporation assay[13,17,30,33]. Naïve $CD4^+$ T cells ($1 \times 10^5$ per well) purified from healthy donors were co-cultured with different types of T cells at a ratio of 10:1 in 200 μl of T-cell assay medium containing 2% human AB. After 56 h of culture, [$^3$H]-thymidine was added at a final concentration of 1 μCi per well, followed by an additional 16 h of culture. The incorporation of [$^3$H]-thymidine was measured with a liquid scintillation counter.

**In vivo studies**. NOD-scid $IL2R\gamma^{null}$ mice (NSG, Stock no. 005557, strain NOD.Cg-Prkdc$^{scid}$Il2rg$^{tm1Wjl}$/SzJ) were purchased from The Jackson Laboratory and maintained in the institutional animal BSL2 facility. Both male and female NSG

mice with 6–8 weeks were used for the studies. All animal studies have been approved by the Institutional Animal Care Committee at Saint Louis University (protocol No. 2411). Naive CD8$^+$ T cells (5 × 10$^6$ per mouse), CD4$^+$CD25$^{hi}$ Treg (2 × 10$^6$ per mouse), and CD4$^+$CD25$^-$effector T cells (2 × 10$^6$ per mouse) were pre-activated with anti-CD3 (2 μg/ml) and adoptively co-transferred into NSG mice through intravenous injection into the following groups: naive CD8$^+$ T cells alone, naive CD8$^+$ T cells plus Treg cells, or naive CD4$^+$ T cells plus CD4$^+$CD25$^-$ effector T cells. Five to 10 mice were included in each group. In a parallel experiment, anti-CD3-activated naive CD8$^+$ T cells were pretreated with or without ATM inhibitor KU55933 (10 μM), STAT1 inhibitor MTA (5 μM), or STAT3 inhibitor S3I201 (10 μM) for 24 h prior to adoptively transfer into the mice. Blood, LNs, and spleens (SP) were collected at 12 days post injection. The transferred human CD4$^+$ and CD8$^+$ T cells were isolated by antibody-coated microbeads (Stemcell) for subsequent phenotypic and functional analyses in vitro. SA-β-Gal staining, flow cytometry analyses, and $^3$H-thymidine incorporation assays were performed as described in above studies.

**Statistical analysis**. Unless indicated otherwise, data are reported as mean ± SD from at least three experiments with similar results. Statistical analysis was performed using the GraphPad Prism5 software. The one-way ANOVA was used for multiple group comparisons. The paired Student's $t$ test was used for a single comparison between two groups, and the nonparametric $t$ test was also chosen if the sample size was too small and not fit Gaussian distribution.

**Data availability**. The authors declare that the data supporting the findings of this study are available within the article and its supplementary information files, or are available from the corresponding author upon request. Microarray data that support the findings of this study have been deposited in Gene Expression Omibus with the primary accession code GSE38765.

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

## Acknowledgements

We thank Joy Eslick and Sherri Koehm for FACS sorting and analyses. We thank Drs Seth Crosby and Qunyuan Zhang at Washington University in St. Louis for performing microarray analyses. This work was partially supported by grants from the American Cancer Society (RSG-10-160-01-LIB, to G.P.), Melanoma Research Alliance (to G.P.), and the NIH (AI097852, AI094478, and CA184379 to G.P.).

## Author contributions

X.L., W.M. and G.P.: designed the research, analyzed the data, prepared figures, and wrote the paper. X.L., W.M., J.Y., L.L. and Y.Z.: performed the experiments. E.C.H.: provided tumor samples and clinical information. D.F.H.: advised the design of research and discussed the manuscript.

## Additional information

**Competing interests:** The authors declare no competing financial interests.

