## [Peer Review File · Nature Communications]

Reviewers' comments:

Reviewer #1

(Remarks to the Author):

This manuscript by Mo et al. is focused on the study of T cell senescence in the presence of physiological regulatory T cells (Treg) and tumor derived $\gamma\delta$ Treg cells. Although this phenomenon was already described by the same group in two previous papers, Mo et al. in this manuscript further investigate the molecular mechanisms involved in human Treg-mediated T cell senescence: competition between Treg and T cells causes DNA damage and the activation of the DNA damage response, particularly the ATM kinase. This activation causes senescence induction through the cooperation between the MAPK ERK1/2 and STAT1/STAT3 pathways. Blocking ATM activity in a murine model results in the inhibition of Treg-mediated T cell senescence.

Overall the manuscript is well written and the results look solid, but I feel that it can be improved in several parts:

Major points

A) In panel 1A and B, quantification by flow cytometry of DDR proteins was performed by studying only their phosphorylated versions. Given that the treatment were performed for several days, a control using antibodies against total protein would exclude that the increase is due to a increased expression of the protein, rather than an increased phosphorylation.

B) Panel 5E is described in the manuscript as the image of a ImmunoFISH experiment: as far as I can see this is not an immunoFISH, but an immunofluorescence experiment.

C) When describing immunofluorescence experiments, the term "co-localization" is used in correlation with the observation that the markers appear in the same cells upon stimulation with nTreg. Since the signal for all the markers tested is pan-nuclear, this term could be misleading. We suggest to use a different term, such as "double positive". Regarding this, for all the experiments only one representative images of each condition is shown. It would be better to also include a quantification of the markers signal. Moreover, on the slide two cell types are present: naive CD4 and the cells used for the treatment. Is it possible to include a marker to distinguish the cell type?

D) In figure 3A and B, the authors compare the amount of glucose transporters Glut1/Glut3 mRNAs and the rate of glucose intake between effectors T cells (CD4+CD25-, nTreg and $\gamma\delta$ Treg). In the manuscript however, their working hypothesis is that the competition for glucose ultimately causing the senescence is between the responder T cell and Treg cells, not between T effectors. In order to support this model, quantification of Glut1/Glut3 and glucose intake in naive CD4+ and CD8+ cells is advisable. Moreover, in Figure 3E the experiment is performed only using nTreg cells. What about $\gamma\delta$ Treg?

E) In Figure 5C the ERK1/2 inhibitors U0126 and SB203580 are used. I would add as a positive control a blot for pERK1/2. In Figure 5D the authors claim that upon KU5593 treatment pSTAT1 in pSTAT3 signal is decreased: for pSTAT3 it doesn't seem the case.

F) Describing panel 6A, the authors claim that PD-1 expression does not change. I would say that there is a mild increase, comparable to CD160.

G) In Figure 6B the expression of different pro-inflammatory factors are evaluated at different time points (4h, 8h and 24h) following effectors T cell treatment. Why use this short time scale, as every experiment is evaluated at 3 and 5 days?

H) Describing Figure 8D, the authors write that CD8+ cells co-transfected with Treg cells "potently suppress the proliferation of responding CD4+". To me the effect is mild.

I) The authors very thoroughly dissect the senescence pathway downstream ATM and identify glucose competition as a possible reason of DNA damage triggering. But no experimental evidence or explanation in the discussion is given on how glucose competition can be a cause of DNA damage.

Minor points:

- A) The flow of the manuscript describing figure 1 and 2 is weird: panels from figure 2 are presented before the last panel of figure 1. I suggest to change the panel disposition.
- B) In co-culture experiments panels, the indication of the ratio of naive CD4+ or CD8+ cells over nTreg or CD4+CD25- cells in co-culture is not always indicated. I suggest to either write in all panels the culture ratio, or clarify the reason why in certain experiments a different ratio was used.
- C) When describing immunofluorescence experiments, the authors don't mention for how many days the cells were treated with nTreg and CD4+CD25- cells. Please clarify.
- D) In panel 2A, the experiment is carried out only for 3 days instead of 3-5 days of the experiment in panel 1A/B. For consistency reasons, it would be better to show also the 5 days time point.
- E) When introducing JAK inhibitor Tyrphostin AG490 (page 10, line 16), I would mention that JAK is upstream in the STAT1/STAT3 pathway.
- F) I suggest to put the factors listed in supplemental panel S3 in the same order as figure 6A for clarity.
- G) Consistency in the figures: Treg cells are sometimes called nTreg and sometimes Treg

Reviewer #2

(Remarks to the Author):

This study by Mo et al, investigated the mechanisms through which regulatory T cells and gd T cells isolated from tumours promote CD4+CD25- T cells to become senescent. The data shows that when responder CD4+CD25- T cells were activated by CD3 crosslinking in the presence of Tregs or gamma/delate (gd) T cells (purified from breast tumours) upregulated DNA damage signalling pathways as seen by increased phosphorylation of ATM, H2AX, CHK2 and p53BP. The induction of T cell senescence was reduced when this signalling pathway was inhibited using a ATM inhibitor. The authors then present data suggesting that nTregs and breast tumour derived gd T cells induce this DNA damage signalling pathway and cellular senescence through competition for glucose. The authors then present data implicating STAT1 and STAT3 signalling in the senescence response. Finally, the authors demonstrate that senescent T cells induced in their system are not functionally exhausted and are distinct from anergic T cells.

Main concern:

(1) The authors argue that senescence is induced due to the ability of nTregs and tumour derived gd T cells to compete for glucose and therefore deprive CD4+CD25- responder T cells of sufficient glucose, which somehow leads to an increased DNA damage response. However, this is not proven.

To prove that Tregs and gd T cells are inducing senescence by depriving the CD4+CD25- responder T cells of glucose the authors would need to inhibit glucose uptake specifically into the Tregs or gd T cells in the co-cultures and show that senescence does not occur. In the experiments they show in Fig 3C and 3D they have treated Tregs/gd for 1 day with a Glucose transporter inhibitor or 2DG and then co-culturing with CD4+CD25- T cells (presumably without these inhibitors - not completely clear in manuscript). Therefore, glucose uptake into the Tregs/gd T cells is not blocked during the coculture with responder T cells. This will certainly be the case for phloretin, which is a competitive inhibitor of glucose transporters and once this inhibitor is washed out then glucose transport can continue.

The data in Fig 3A-D suggests that nTregs and gd have increased levels of glycolysis compared to the CD4+CD25- T cells that is important for the ability to induce senescence in T cells. Inhibition of glucose uptake or glycolysis for one day appears to be altering the function of nTregs and gd T cells such that when they are then co-cultured with CD4+CD25- responder T cells they cannot

induce senescence. However, the data does not support the proposed mechanism that nTregs compete for glucose to induced DNA damage signalling in the responder T cells.

It is not very clear what ratios the cells were co-cultured at each experiment in this study. I took it that the cocultures were 10 CD4+CD25- T cells to 1 Treg/ gd . It is not plausible that 1 Treg/ can deprive 10 CD4+CD25- responder T cells of glucose.

In Figure 3E and 3F high dose of glucose (25 mM) are added to the coculture and are observed to reduce the induction of DNA damage signalling and T cell senescence. The authors need to measure glucose levels in the co-culture experiment on day 1 – 5. It seems likely that glucose is becoming depleted globally in these cocultures when 10 mM glucose is used (Fig. 1-2) and adding 2.5-3.5 times more glucose (Fig. 3E-F) simply ensures that there is sufficient glucose for the duration of the assay. While this data does appear to link glucose levels to the induction of ATM signalling and T cell senescence, it does not support the authors argument that nTregs deprive CD4+CD25- T cells of glucose. Perhaps low glucose levels are facilitating the DNA damage response in responder T cells that is initiated by a soluble factor from the nTreg or a cell-cell interaction with the nTreg. Measuring the glucose levels in the media of these cultures would shed some light on this matter.

I would also question whether nTregs depriving responder T cells of glucose could ever be a relevant mechanism in vivo within a tumour. Tumour cells are highly glycolytic and use large amounts of glucose to the point that they can largely deplete glucose from the tumour microenvironment. In this context, the contribution of the glucose used by nTregs or gd to global glucose depletion in the tumour microenvironment will be negligible.

Other points:

Figure 1:

Authors should show that pCHK2, pH2AX and p53BP signals are lost when cells are treated with KU99533. This would provide confidence in these flow cytometry assays; negative controls are important to validate flow cytometry based phospho-protein analysis.

While the levels of CD27 and CD28 do increase with KU55933, they are not restored to control levels. Have the authors considered if the DNA damage sensing kinase ATR is also activated in these cells? If pH2AX levels are only partially reduced by KU99533, this might suggest that ATR is also active in these cells, as both ATM and ATR phosphorylate H2AX.

Figure 1E-F. Increased p53, P16 and P21 correlating with pATM, pH2AX, and pCHK2 is not really surprising. These events are all part of the established DNA damage response. Taken together the data is convincing that the DNA damage response is activated in these CD4 T cells.

Figure 1C, statistical analysis is not clear. Ku55933 appears to partially reduce SA-gal +ve cells. Is KU55933 statistically different to the CD4+CD25- control group?

Supplementary Figure 1A – There must be an error in this figure. The figures show a cell gate and a figure that is presumably the % positive cells within this gate. But in some instances (p-CHK2, left panel) the gate includes the main peak of cells but the % positive is stated as 11%. While the differences between the different conditions look real, the % positive figures in this figure cannot be correct.

Figure 2:

Figures are not in order. Figure 2A is before Figure 1E.

Figure 2A: Same point as for Supplementary Figure 1A above. The gates and % positive figures cannot be correct. See p-CHK2 left panel where the gate encompasses most of the single peak of cells and is labelled as 10.9 % positive. This cannot be correct. Many examples of this in this figure.

Figure 2B: Again, the response is not completely blocked with KU88933. Authors should include statistical analysis of control versus KU99533. Is there a role for ATR?

Reviewer #3

(Remarks to the Author):

In this paper, Mo and colleagues identify novel molecular mechanisms responsible for human nTreg-mediated suppression and provide evidence that in vitro and in vivo targeting of DNA damage response or STAT1/3 signaling can reverse nTreg-induced suppression. The experimental design involves a series of co-incubation experiments in which human CD4⁺CD25^{hi} cells sorted from PBMC of normal donors are co-incubated with CD4⁺ or CD8⁺ T effector (Teff) and the mechanisms responsible for suppression they induce are evaluated. The authors first demonstrate that these nTreg induce senescence in Teff via activation of nuclear kinase ATM-associated DNA damage response. The authors then define senescence as a phenotypically and functionally characteristic T-cell type, show that senescence is a result of a competition for glucose and that STAT1/3 regulate cellular senescence, which also involves cooperation with the MAPK and Erk1/2 pathway in responder Teff. A series of experiments involving loss-of-function and gain-of-function scenarios are then presented to show that senescent T cells are functional, mediate suppression and are different both phenotypically and functionally from anergic T cells and exhausted T cells. Finally, the authors use NSD immunocompetent mice, to which they transfer human nTreg and Teff, to show in vivo that the ATM-induced DNA damage response is the main cause of suppression mediated by human Treg.

The manuscript comes from an experienced group of investigators, and the presented experiments are expertly performed. The problems arise, however, with interpretation of the presented experiments. There are multiple conceptual concerns, and biological significance the results is overstated.

If the authors limited their intent to demonstrating that in vitro generated nTreg induce senescence by ATM-associated DNA damage response and that this mechanism is utilized by human nTreg to suppress Teff functionally, then they would have accomplished the objective. Whether this mechanism of suppression is relevant to in vivo interactions between Treg and Teff remains unclear. The in vivo experiments in immunoincompetent NSG mice re-populated with human nTreg and T eff are not a convincing model, because cellular interactions occur in the artificial environment where human cells find themselves interacting with mouse tissue cells and mouse residual immune cells.

The authors assume that normal human T cells can be categorized into senescent, exhausted and anergic T cell subtypes based on their distinct phenotypic and in part functional attributes. The authors go as far as to state that nTreg inducing senescence via ATM-associated DNA damage response represent a distinct lineage of T cells. There is no evidence for this, nor is there convincing evidence for the existence under normal physiological conditions of the three T-cell categories, as a considerable degree of overlap is evident.

The authors are working with nTreg isolated by sorting of CD4⁺CD25^{hi} T cells. This is a highly

biased approach to the isolation of Treg. What is the complete phenotype of the isolated nTreg? Are highly activated CD4+ cells selected as well as nTreg?? These isolated nTreg are then placed in long-term co-cultures with Teff and plenty of possibilities for selective interactions, given the well-known plasticity of human T cells. So the cellular origin and unique features of the nTreg inducing senescence in Teff are open to criticism.

The authors are working with nTreg derived from normal PBMC. Yet, the text is peppered with comments to tumor-reactive T cells and tumor-associated Treg. Based on the presented data no comments relevant to T cells in cancer or those in human tumors can be made. Similarly, references to breast cancer-derived γ/δ Treg made in the text bear no relationship to the presented data. Treg subsets in cancer patients or those in patients with chronic infections are not comparable to nTreg populating PBMC of normal donors.

References are made in the text to tumor-induced suppression of Teff by mechanisms involving cAMP, and the authors attempted to show using inhibitors of adenylyl cyclase or PKA that T cell senescence was not dependent on this mechanism of suppression. As there are obviously many different mechanisms Treg can and do use for suppression, depending on the environment in which these cells find themselves, it is not surprising that in vitro induced and measured suppression by nTreg depends on the DNA damage response.

Overall, while the possibility that human nTreg generated in vitro are able to mediate suppression via ATM-associated DNA damage response is not questioned, the biological significance of this type of regulation operating in vivo remains unclear. The attempt to treat nTreg that use this mechanism of suppression in vitro as a unique subset, or a distinct lineage, of Treg and to compare or contrast the senescent-inducing nTreg to those inducing exhaustion or anergy is not supported by the presented data. The categories of Treg and mechanisms of suppression these cells utilize are context dependent and likely vary broadly between health and disease.

The manuscript contains numerous grammatical errors which should be corrected.

Reviewer #1

Overall the manuscript is well written and the results look solid

Response: We appreciate the positive comments and very helpful suggestions from this reviewer for improving our manuscript.

Major points

1. In panel 1A and B, quantification by flow cytometry of DDR proteins was performed by studying only their phosphorylated versions. Given that the treatment were performed for several days, a control using antibodies against total protein would exclude that the increase is due to a increased expression of the protein, rather than an increased phosphorylation.

Response: I agree with this reviewer's concern. As suggested by this reviewer, we have repeated the studies shown in the previous Figure 1A and 1B, and further shown that treatment with both nTreg and $\gamma\delta$ Treg cells induced significant activation and phosphorylation of DNA damage molecules ATM, CHK2, H2AX and 53BP1 in responder CD4⁺ and CD8⁺ T cells (**new Figure 1** in this revised manuscript). Furthermore, we confirmed that the increased phosphorylation levels of those DNA damage molecules were not due to the increased their total protein levels in senescent T cells induced by human Treg cells (**Supplemental Figure S1** in this revised manuscript).

These new data have been now included in **Figure 1** and **Supplemental Figure S1A**, and described in **page 6, Lines 21-24**, and in the **Materials and Method**, and **Figure legends** sections in this revised manuscript.

2. Panel 5E is described in the manuscript as the image of a ImmunoFISH experiment: as far as I can see this is not an immunoFISH, but an immunofluorescence experiment.

Response: I apologize for the confusion. We have corrected it in this revised manuscript (**Page 13, lines 3-10**).

3. When describing immunofluorescence experiments, the term “co-localization” is used in correlation with the observation that the markers appear in the same cells upon stimulation with nTreg. Since the signal for all the markers tested is pan-nuclear, this term could be misleading. We suggest to use a different term, such as “double positive”. Regarding this, for all the experiments only one representative images of each condition is shown. It would be better to also include a quantification of the markers signal. Moreover, on the slide two cell types are present: naive CD4 and the cells used for the treatment. Is it possible to include a marker to distinguish the cell type?

Response: Thanks for this reviewer's suggestions. I have changed the term “co-localization” to “co-expression”.

In addition, I understand the reviewer's concern about the quantification of the markers. Actually, all the markers shown in the immune-fluorescence studies for senescent T cells in this manuscript have been quantified either by western-blot or flow cytometry analyses in the current studies or our previous publications^{1,2} (Ye, et al., Blood 2012 and Ye, et al., J Immunol 2013).

By the way, I apologize for the confusion. All the responder T cells for the studies in this manuscript have been sorted by FACS from the co-cultures with Treg cells based on the CFSE labeling strategy (See **Material and Method** section, **page 25, line 1-3**). Therefore, we do not need to distinguish them with Treg cells.

4. In figure 3A and B, the authors compare the amount of glucose transporters Glut1/Glut3 mRNAs and the rate of glucose intake between effectors T cells (CD4+CD25-, nTreg and $\gamma\delta$ Treg). In the manuscript however, their working hypothesis is that the competition for glucose ultimately causing the senescence is between the responder T cell and Treg cells, not between T effectors. In order to support this model, quantification of Glut1/Glut3 and glucose intake in

naive CD4⁺ and CD8⁺ cells is advisable. Moreover, in Figure 3E the experiment is performed only using nTreg cells. What about $\gamma\delta$ Treg?

Response: As suggested by this reviewer, we have performed additional experiments. We found that expression levels of levels of Glut1 and Glut3 in naïve CD4⁺ and CD8⁺ T cells are very similar as CD4⁺CD25⁻ effector T cells (Data not shown). In addition, we found that glucose uptake ability of naïve CD4⁺ and CD8⁺ T cells is much lower than that of nTreg and tumor-derived $\gamma\delta$ Treg cells, using a 2-NBDG labeling assay (**Figure 3B** in this revised manuscript).

To further support the hypothesis that the competition for glucose ultimately causing the senescence is between the responder T cell and Treg cells, not between T effectors, we also determined the glucose concentrations in the culture medium Treg, naïve T cells, and Treg-naïve T co-cultures. We further confirmed that nTreg highly consumed glucose in the medium during 3 days of culture compared with that of naïve CD4⁺ T cells, indicating a heightened metabolic activity. In addition, co-culture with nTreg cells but not control CD4⁺CD25⁻ effector T cells significantly decreased the glucose level in the culture medium (**Figure 3C** in this revised manuscript). These results collectively suggest that human Treg cells have heightened glucose consumption distinct from effector T cells.

In addition, as suggested by this reviewer, we have also included the new studies showing that addition of high concentrations of glucose (25 and 35 mM) dramatically prevented the responder T cell senescence mediated by $\gamma\delta$ Treg cells (**Figure 3H** in this revised manuscript).

These new data have been now included in **Figure 3B, 3C and 3H**, and described in **page 9, Lines 8-18** and **page 10, lines 7-14**, and in the Materials and Method section (**page 25**) in this revised manuscript.

5. In Figure 5C, the ERK1/2 inhibitors U0126 and SB203580 are used. I would add as a positive control a blot for pERK1/2. In Figure 5D the authors claim that upon KU5593 treatment pSTAT1 in pSTAT3 signal is decreased: for pSTAT3 it doesn't seem the case.

Response: Based on this reviewer's suggestions, we have repeated the experiments, which have now been shown in **Figure 5C** and **5D** with all the controls. In addition, our newly generated results have clearly shown that blockage of ATM signaling with KU55933 significantly decreased STAT1 and STAT3 phosphorylation and activation in Treg-induced senescent T cells (**Figure 5D** in this revised manuscript).

These new data have been now included in **Figure 5C** and **5D** in this revised manuscript.

6. Describing panel 6A, the authors claim that PD-1 expression does not change. I would say that there is a mild increase, comparable to CD160.

Response: I have corrected it in this revised manuscript (**Page 13, lines 23-25**).

7) In Figure 6B the expression of different pro-inflammatory factors are evaluated at different time points (4h, 8h and 24h) following effector T cell treatment. Why use this short time scale, as every experiment is evaluated at 3 and 5 days?

Response: Based on our experience, the mRNA expression level changes for cytokines occur much earlier than their protein level alterations. Therefore, we evaluated the mRNA expression of a panel of pre-inflammatory and effector cytokines at early time points (4h, 8h and 24h) following Treg treatment. In addition, those cytokine mRNA expression results have been confirmed at the protein levels in senescent T cells induced by both nTreg and $\gamma\delta$ Treg cells at day 3 and day 5 in our previous studies (Ye, Blood 2012 and Ye, J Immunol 2013).

8) Describing Figure 8D, the authors write that CD8+ cells co-transfected with Treg cells “potently suppress the proliferation of responding CD4+”. To me the effect is mild.

Response: We have changed it based on this reviewer’s suggestion.

9) The authors very thoroughly dissect the senescence pathway downstream ATM and identify glucose competition as a possible reason of DNA damage triggering. But no experimental evidence or explanation in the discussion is given on how glucose competition can be a cause of DNA damage.

Response: I understand this reviewer’s concern. To address this concern, we have performed additional experiments and identify the causative link between glucose shortage and T cell senescence. Recent studies have shown that AMPK is an important energy sensor activated by metabolic stresses³. AMPK also directly regulates DNA-damage and T cell senescence^{4,5}. As expected, our data have demonstrated that significantly increased phosphorylation of AMPK in responder T cells was induced by Treg cells, indicating the molecular link between glucose shortage and T cell senescence (**Figure 3D** in this revised manuscript). Furthermore, phosphorylated-AMPK was also promoted in T cells with low concentrations of glucose (**Figure 3E** in this manuscript). **Importantly, our newly generated data clearly suggested** that deprivation of glucose in co-culture medium directly triggers T cell senescence and DNA damage response (**Attached Figure 1** and see **response for Reviewer 2, question #1**). In addition, using loss-of-function and gain-of-function studies, we further identified that heightened consumption of glucose mediated by Treg cells is the cause for ATM-induced DNA damage response and induction of T cell senescence in responder T cells mediated by human Treg cells (**Figure 3F-3I** in this manuscript). These studies collectively identify the causative link between Treg metabolism and the molecular process on responder T cell fate and function mediated by glucose competition/deprivation during their cross-talk and interactions.

These new data have been now included in **Figure 3D-3E**, and described in **page 9, lines 18-25**, and in the Discussion section (**page 18, lines 25-26; page 19, lines 1-11**) in this revised manuscript.

Minor points:

A) The flow of the manuscript describing figure 1 and 2 is weird: panels from figure 2 are presented before the last panel of figure 1. I suggest to change the panel disposition.

Response: We have reorganized the figures originally shown in the Figures 1 and 2, and the supplemental Figure 1 in this revised manuscript.

B) In co-culture experiments panels, the indication of the ratio of naive CD4⁺ or CD8⁺ cells over nTreg or CD4⁺CD25⁻ cells in co-culture is not always indicated. I suggest either write in all panels the culture ratio, or clarify the reason why in certain experiments a different ration was used.

Response: I apologize for the confusion. The ratio of naïve T cells co-cultured with Treg cells or control effector CD4⁺CD25⁻ T cells is 4:1 in all the studies in this manuscript except the [³H]-thymidine incorporation assay with 10:1 ratio. We have clarified it in **the Material and Method** section in the manuscript (**pages 23-26**).

C) When describing immunofluorescence experiments, the authors don't mention for how many days the cells were treated with nTreg and CD4⁺CD25⁻ cells. Please clarify.

Response: For the immunofluorescence experiments, naive CD4⁺ T cells or CD8⁺ T cells were co-cultured with Treg or control T cells labeled with CFSE (4.5 μM), at a ratio of 4:1 in anti-CD3-coated 24-well plates for 3 days. I have clarified it in Material and Method section in this revised manuscript (**Page 25, line 1-3**).

D) In panel 2A, the experiment is carried out only for 3 days instead of 3-5 days of the experiment in panel 1A/B. For consistency reasons, it would be better to show also the 5 days time point.

Response: In order to keep the entire studies consistence, we have deleted the results for 5 days and kept all the data for 3-day culture in Figures 1 and 2, and supplemental Figures S1 and S2 in this revised manuscript.

E) When introducing JAK inhibitor Tyrphostin AG490 (page 10, line 16), I would mention that JAK is upstream in the STAT1/STAT3 pathway.

Response: As suggested by this reviewer, we have added the senescence in the revised manuscript (**page 11, lines 12-13**).

F) I suggest to put the factors listed in supplemental panel S3 in the same order as figure 6A for clarity.

Response: It has been corrected, as this reviewer suggested.

G) Consistency in the figures: Treg cells are sometimes called nTreg and sometimes Treg.

Response: Thanks for the suggestion. We have already changed it in this revision.

Reviewer #2

I would like to thank this reviewer again for his/her insightful suggestions to improve our manuscript.

Main concern:

1) The authors argue that senescence is induced due to the ability of nTregs and tumour derived gd T cells to compete for glucose and therefore deprive CD4+CD25- responder T cells of sufficient glucose, which somehow leads to an increased DNA damage response. However, this is not proven.

Response: I understand this reviewer's concern. Our group has been working on the tumor-associated Treg cells and tumor immunity during the past 15 years. So far, the suppressive mechanism mediated by human Treg cells is unknown. Our current studies provide the first evidence that induction of T cell DNA-damage response and senescence is a novel and important mechanism mediated by human Treg cells. Furthermore, we identified that that glucose competition between Treg and responder T cells could be the cause for triggering the cell senescence and DNA damage in responder T cells.

To address this reviewer's concern, we have performed additional experiments and provided our recently generated new data to further support this novel mechanism and conclusions in the following ways:

(1) Activated nTreg and tumor-associated Treg cells display strong desire for glucose metabolism compared with effector T cells.

We found that both nTreg and $\gamma\delta$ Treg cells have much higher expression levels of Glut1 and Glut3, compared with effector T cells (**Figure 3A** in the revised manuscript). Furthermore, our results clearly showed that nTreg and tumor-derived $\gamma\delta$ Treg cells have a significantly higher glucose uptake compared with control naïve CD4⁺ and CD8⁺ T cells, as well as CD4⁺CD25⁻ effector T cells using a 2-NBDG assay (**Figure 3B** in the revised manuscript). We then confirmed that nTreg highly consumed glucose in the medium during 3 days of culture compared with naïve CD4⁺ T cells. In addition, co-culture with nTreg cells but not control CD4⁺CD25⁻ effector T cells significantly decreased the glucose level in the culture medium (**Figure 3C** in the revised manuscript). These results clearly indicate a heightened metabolic activity of Treg cells.

(2) Activation of AMPK in responder T cells induced by Treg cells, inducing T cell DNA-damage response and senescence.

AMP-activated protein kinase (AMPK), an important nutrient and energy sensor, is activated by metabolic stresses³. Recent studies suggest that AMPK is also activated by DNA-damage agents directly regulating T cell senescence^{4,5}. To determine the causative link between glucose shortage and T cell senescence, we observed significantly increased phosphorylation of AMPK in responder T cells treated with Treg cells, but not with control T cells (**Figure 3D** in the revised manuscript). Furthermore, phosphorylated-AMPK was also promoted in T cells with low concentrations of glucose (**Figure 3E** in the revised manuscript). These results strongly suggest that Treg-induced T cell senescence is regulated by the glucose metabolic stress.

(3) Deprivation of glucose directly triggers T cell senescence and suppression.

To further test our hypothesis that the lack of enough glucose highly consumed by Treg cells may result in responder T cell senescence, we investigated whether low concentrations of glucose can directly induce T cell senescence. Anti-CD3-activated naïve CD4⁺ T cells were cultured with different low concentrations of glucose (0-11 mM) for 5 days and then studied for cell senescence. As expected, we found that the increased senescent CD4⁺ T cell populations were markedly induced in the medium with low concentrations of glucose in a dose-dependent manner (**Attached Figure 1A**). Furthermore, low concentrations of glucose in the medium also significantly down-regulated co-stimulatory molecules CD27 and CD28, and promoted the expression of cell cycle regulated genes p53, p21, and p16 (**Attached Figure 1B & 1C**), further suggesting that shortage of glucose for T cells drove development of senescent T cells. To explore the mechanisms potentially involved in the induction of cellular senescence mediated by low concentrations of glucose, we determined the induction of DNA damage in responder T cells. We found that CD4⁺ T cells co-cultured with normal concentration of glucose (11 mM) only had minor expression of phosphorylated ATM. In contrast, low glucose treatment significantly induced phosphorylation of ATM in co-cultured CD4⁺ T cells (**Attached Figure 1D**). We also observed significant activation and phosphorylation of the other key DNA damage-associated molecules H2AX, 53BP1, and CHK2 in CD4⁺ T cells treated with low concentrations of glucose (**Attached Figure 1D**). To further confirm that induction of ATM-associated DNA damage is the cause for low glucose-induced T cell senescence, we next determined whether we can prevent the senescence induction in T cells through functional blockage of ATM-induced DNA damage in T cells using the loss-of-function approach with the ATM specific pharmacological inhibitor KU55933. Pre-treatment with KU55933 in responder T cells dramatically prevented the induction of SA-β-Gal⁺ T cell populations in CD4⁺ T cells induced by low glucose (**Attached Figure 1E**). **These further results clearly support our conclusion that Treg-mediated heightened glucose consumption triggers cell senescence and suppression in responder T cells during their cross-talk.**

Attached Figure 1. Treg-mediated competition of glucose consumption with responder T cells is the cause for cell senescence and suppression in responder T cells.

(A) Significantly increased SA-β-Gal⁺ T cell populations were induced in naïve CD4⁺ T cells cultured in the medium with low concentrations of glucose. Anti-CD3-activated naïve CD4⁺ T cells were cultured with different concentrations of low glucose for 5 days. Normal medium with 11 mM glucose served as a control. The CD4⁺ T cells were determined for cell senescence with SA-β-Gal expression analyses. (B) Down-regulation of CD28 and CD27 in naïve CD4⁺ T cells induced by the culture medium with low concentrations of glucose. Cell treatment and procedure were the same as in (A). CD27 and CD28 expression in treated naïve CD4⁺ T cells were analyzed by the flow cytometry. Mean

fluorescence intensity (MFI) was calculated. (C) Expression of cell cycle regulated genes p53, p21, and p16 in CD4⁺ T cells cultured in low glucose medium. Cell treatment and procedure were the same as in (A). p53, p21, and p16 expression in treated naïve CD4⁺ T cells were analyzed by the flow cytometry. Mean

Fluorescence Intensity (MFI) was calculated as shown in the right. **(C)** Increased expression of cell cycle regulatory molecules p53, p21 and p16 in naïve CD4⁺ T cells cultured with low concentrations of glucose medium. Cell treatment and procedure were the same as in (A). The p53, p21 and p16 expression in treated CD4⁺ T cells were analyzed by the flow cytometry. **(D)** Phosphorylated activation of ATM and other associated molecules H2AX, 53BP1 and CHK2 in naïve CD4⁺ T cells cultured with low concentrations of glucose medium. Cell treatment and procedure were the same as in (A). The p-ATM, p-H2AX, p-53BP1, and p-CHK2 expression in cultured naïve CD4⁺ T cells were analyzed by the flow cytometry. **(E)** Treatment of T cells with an ATM specific inhibitor KU55933 significantly prevented CD4⁺ T-cell senescence induced by the low glucose condition. Anti-CD3 pre-activated naïve CD4⁺ T cells were cultured with different concentrations of glucose medium in the presence or absence of ATM specific inhibitor KU55933 (20 µM) for 5 days. SA-β-Gal expression in CD4⁺ T cells was determined with SA-β-Gal staining. Data shown are mean ± SD from three independent experiments. **p<0.01, compared with the medium only group.

(4) Using the loss-of-function and gain-of-function strategies to confirm that glucose competition between Treg and responder effector T cells triggers cell senescence and DNA damage in responder T cells during their interactions.

To further determine whether the competition of glucose consumption between Treg and responder T cells is the main reason for Treg-induced T cell senescence and DNA damage response, We found that blockages of both glucose transporter and glycolysis with respective inhibitors phloretin and 2-deoxy-D-glucose (2-DG), can significantly reverse Treg suppressive effect on the proliferation of responder T cells and markedly decrease responder T cell senescence induced by both nTreg and breast tumor-derived γδ cells (**Figure 3F & 3G** in this revised manuscript). Using the gain-of-function assays, we found that addition of high concentrations of glucose (25 and 35 mM) dramatically prevented the responder T cell senescence and down-regulated the phosphorylated activation of ATM, and the downstream molecules H2AX and 53BP1 and CHK2 as well in responder T cells mediated by nTreg cells (**Figures 3H and 3I** in this revised manuscript).

Collectively, these results clearly indicate that glucose competition between Treg and responder effector T cells triggers cell senescence and DNA damage in responder T cells during their interactions. These new data have been now included in **Figure 3C-3E**, and described in **page 9, Lines 8-25**, and in the Discussion section (**page 18, lines 25-26; page 19, lines 1-11**) in this revised manuscript.

2) To prove that Tregs and gd T cells are inducing senescence by depriving the CD4+CD25-responder T cells of glucose the authors would need to inhibit glucose uptake specifically into the Tregs or gd T cells in the co-cultures and show that senescence does not occur. In the experiments they show in Fig 3C and 3D they have treated Tregs/gd for 1 day with a Glucose transporter inhibitor or 2DG and then co-culturing with CD4+CD25- T cells (presumably without these inhibitors - not completely clear in manuscript). Therefore, glucose uptake into the Tregs/gd T cells is not blocked during the coculture with responder T cells. This will certainly be the case for phloretin, which is a competitive inhibitor of glucose transporters and once this inhibitor is washed out then glucose transport can continue.

Response: We apologize for the confusion. The inhibitors in the experiments of this manuscript are utilized both in the pretreatment of Treg cells and in the co-cultures as well. The screening for the concentrations of the glucose metabolism inhibitors for co-cultures is based on the effects

on Treg and effector T cells, which can suppress the glucose metabolism/function in Tregs, but not inhibit responder T cell proliferation or induce responder T cell senescence. We have clarified it in the material and method section (**page 23, lines 20-24**) in this revised manuscript.

3) The data in Fig 3A-D suggests that nTregs and gd have increased levels of glycolysis compared to the CD4+CD25- T cells that is important for the ability to induce senescence in T cells. Inhibition of glucose uptake or glycolysis for one day appears to be altering the function of nTregs and gd T cells such that when they are then co-cultured with CD4+CD25- responder T cells they cannot induce senescence. However, the data does not support the proposed mechanism that nTregs compete for glucose to induced DNA damage signaling in the responder T cells.

Response: I do understand this reviewer's concern. **Please see the response to major concern #1 above for this Reviewer.** Our newly generated data combined with the previous results clearly indicate that glucose competition between Treg and responder effector T cells triggers cell senescence and DNA damage in responder T cells during their interactions.

4) It is not very clear what ratios the cells were co-cultured at each experiment in this study. I took it that the cocultures were 10 CD4+CD25- T cells to 1 Treg/ gd . It is not plausible that 1 Treg/ can deprive 10 CD4+CD25- responder T cells of glucose.

Response: The ratio of naïve T cells co-cultured with Treg cells or control effector CD4+CD25- T cells is 4:1 in all the studies in this manuscript except the [³H]-thymidine incorporation assay with 10:1 ratio. I have clarified it in the material and method section in the manuscript. In addition, our newly generated data clearly showed that nTreg highly consumed glucose in the medium during 3 days of culture compared with naïve CD4⁺ T cells. Furthermore, co-culture with nTreg cells but not control CD4⁺CD25⁻ effector T cells significantly decreased the glucose level in the culture medium (**Figure 3C** in this revised manuscript), suggesting that human Treg cells have heightened glucose consumption distinct from effector T cells.

5) In Figure 3E and 3F high dose of glucose (25 mM) are added to the coculture and are observed to reduce the induction of DNA damage signalling and T cell senescence. The authors need to measure glucose levels in the co-culture experiment on day 1 – 5. It seems likely that glucose is becoming depleted globally in these cocultures when 10 mM glucose is used (Fig. 1-2) and adding 2.5-3.5 times more glucose (Fig. 3E-F) simply ensures that there is sufficient glucose for the duration of the assay. While this data does appear to link glucose levels to the induction of ATM signaling and T cell senescence, it does not support the authors argument that nTregs deprive CD4+CD25- T cells of glucose. Perhaps low glucose levels are facilitating the DNA damage response in responder T cells that is initiated by a soluble factor from the nTreg or a cell-cell interaction with the nTreg. Measuring the glucose levels in the media of these cultures would shed some light on this matter.

Response: To address this reviewer's concern, our recently generated new data clearly suggest that deprivation of glucose can directly trigger T cell senescence and suppression (**See response to Question #1 for this reviewer and the Attached Figure 1**). In addition, our new data confirmed that nTreg highly consumed glucose in the medium during 3 days of culture compared with naïve CD4⁺ T cells. Furthermore, co-culture with nTreg cells but not control CD4⁺CD25⁻

effector T cells significantly decreased the glucose level in the culture medium (**Figure 3C** in the revised manuscript). Based on the combined new data (**Attached Figures 1 and Figure 3C**), it is clear **that Treg-mediated heightened glucose consumption triggers cell senescence and suppression in responder T cells during their cross-talk**. The results do not likely support the possibility that “low glucose levels are facilitating the DNA damage response in responder T cells that is initiated by a soluble factor from the nTreg or a cell-cell interaction with the nTreg”.

6) I would also question whether nTregs depriving responder T cells of glucose could ever be a relevant mechanism in vivo within a tumour. Tumour cells are highly glycolytic and use large amounts of glucose to the point that they can largely deplete glucose from the tumour microenvironment. In this context, the contribution of the glucose used by nTregs or gd to global glucose depletion in the tumour microenvironment will be negligible.

Response: I acknowledge that we cannot provide the direct evidence to support this novel mechanism and concept *in vivo* in tumor models in this manuscript. However, our previous studies have clearly shown that Treg cells including $\gamma\delta$ Treg cells are the most dominant cell populations in the TILs from breast cancer and melanoma patients (*Immunity* 2004; *Immunity* 2007; *Clin Can Res* 2007; *J Immunol* 2012; *Oncotarget* 2014). In addition, our recent studies have also shown that increase senescent T cell populations in TILs in melanoma and breast cancer patients, as well as in tumor-bearing B16 melanoma and E0771 breast cancer mice (*EMBO Mol Med* 2014, and see the **Response to Reviewer #3’ question # 5, Attached Figure 3 and 4**). Therefore, we do not think that the contribution of the glucose used by nTregs or $\gamma\delta$ Treg to global glucose depletion in the tumour microenvironment is negligible. We agree that tumor cells and tumor-infiltrating T cells (TILs) compete for glucose within the tumor suppressive microenvironment, which is a driver for tumor suppression and progression^{6,7}. Given that promotion of accumulated Treg cells in the tumor microenvironments is a key strategy for malignant tumor to induce immune tolerance, thus our current studies identify a mechanistic link between Treg suppression and T cell senescence in the tumor microenvironment. Our future studies will continue to focus on this novel mechanism and concept for enhanced anti-tumor immunity in different tumor models.

Other points:

1) Authors should show that pCHK2, pH2AX and p53BP signals are lost when cells are treated with KU99533. This would provide confidence in these flow cytometry assays; negative controls are important to validate flow cytometry based phospho-protein analysis.

Response: To address this concern, we have performed additional experiments. We found that treatment with KU55933 in responder T cells can significantly prevent the activation and phosphorylation of ATM, CHK2, H2AX and 53BP1 in responder both CD4⁺ and CD8⁺ T cells treated with nTreg cells (**Supplemental Figure S2A** in this manuscript).

These new data have been now included in supplemental **Figure S2A**, and described in **page 7 Lines 8-10, and in the Supplemental Figure legend** in this revised manuscript.

2) While the levels of CD27 and CD28 do increase with KU55933, they are not restored to control levels. Have the authors considered if the DNA damage sensing kinase ATR is also

activated in these cells? If pH2AX levels are only partially reduced by KU99533, this might suggest that ATR is also active in these cells, as both ATM and ATR phosphorylate H2AX.

Response: I appreciate this reviewer's helpful suggestion. Given that fact that blockage with KU55933 cannot completely prevent T cell DNA damage and senescence induction mediated by Treg cells, we also determined whether the other key regulator ATR also involves regulation of DNA damage response in responder T cells as suggested by this reviewer. We observed that co-culture with nTreg but not control CD4⁺CD25⁻ T cells also promoted the phosphorylation of ATR in responder T cells, suggesting that ATR may involve the regulation of Treg-induced T cell senescence (**Supplemental Figure S3A**). Collectively, these results indicate that initiation of DNA damage is critical and the main cause for the induction of T cell senescence and dysfunction mediated by human Treg cells.

These new data have been now included in supplemental **Figure S3A**, and described in **page 7 Lines 19-23, and in the Supplemental Figure legend** in this revised manuscript.

3) Figure 1E-F. Increased p53, P16 and P21 correlating with pATM, pH2AX, and pCHK2 is not really surprising. These events are all part of the established DNA damage response. Taken together the data is convincing that the DNA damage response is activated in these CD4 T cells.

Response: I agree this comment. Yes, all these studies clearly suggest that initiation of DNA damage is critical and the main cause for the induction of T cell senescence and dysfunction mediated by human Treg cells.

4) Figure 1C, statistical analysis is not clear. Ku55933 appears to partially reduce SA-gal +ve cells. Is KU55933 statistically different to the CD4+CD25- control group?

Response: As suggested, we have performed statistical analysis between the two groups in this revised manuscript.

5) Supplementary Figure 1A – There must be an error in this figure. The figures show a cell gate and a figure that is presumably the % positive cells within this gate. But in some instances (p-CHK2, left panel) the gate includes the main peak of cells but the % positive is stated as 11%. While the differences between the different conditions look real, the % positive figures in this figure cannot be correct.

Response: We apologize for the confusion. We have repeated these studies. The new data are now presented in **Figure 1B** in this revised manuscript.

Figure 2:

6) Figures are not in order. Figure 2A is before Figure 1E.

Response: We have reorganized the figures shown in the original Figures 1 and 2, and the supplemental Figure 1 in this revised manuscript.

7) Figure 2A: Same point as for Supplementary Figure 1A above. The gates and % positive figures cannot be correct. See p-CHK2 left panel where the gate encompasses most of the single peak of cells and is labelled as 10.9 % positive. This cannot be correct. Many examples of this in this figure.

Response: We have corrected the gating and the percentages in this Figure (See **Figure 1C & 1D** in this revised manuscript).

8) Figure 2B: Again, the response is not completely blocked with KU88933. Authors should include statistical analysis of control versus KU99533. Is there a role for ATR?

Response: See Responses for **Other points, Questions 2 &4** above. We have done as suggested.

Reviewer #3

The manuscript comes from an experienced group of investigators, and the presented experiments are expertly performed.

Response: I would like to thank this reviewer for the recognition of our efforts and research work in the field.

Defining the suppressive mechanisms used by human Treg cells is a challenge; although great progress has been made in understudying suppressive mechanisms mediated by mouse Treg cells. We have been trying our best to identify how human Treg interacts with the responder T cells and what are the fate and functional changes of Treg-treated T cells after their cross-talk. Based on our current studies, we have clearly shown that induction of targeted T cell DNA damage response and senescence is the common step in their suppression mediated by different types of Treg cells. In addition, we further dissected the molecular processes for senescence induction mediated by human Treg cells on responder T cells during their interactions in this manuscript.

1) If the authors limited their intent to demonstrating that in vitro generated nTreg induce senescence by ATM-associated DNA damage response and that this mechanism is utilized by human nTreg to suppress T_{eff} functionally, then they would have accomplished the objective. Whether this mechanism of suppression is relevant to in vivo interactions between Treg and T_{eff} remains unclear. The in vivo experiments in immunoincompetent NSG mice re-populated with human nTreg and T_{eff} are not a convincing model, because cellular interactions occur in the artificial environment where human cells find themselves interacting with mouse tissue cells and mouse residual immune cells.

Response: I fully understand this reviewer's concern. Our group has been focusing on the studies to dissect the role of human T cell subsets in anti-tumor immunity and immunotherapy during the past more than 15 years. Humanized mouse models have been widely utilized by our group and other groups as well to study interactions between human T cells and tumor cells in vivo in the research field. We have published serial research papers using the humanized mice to study human T cells, including in *Science* (2005), *Immunity* (2004 and 2007), *Blood* (2012),

Cancer Res (2013), and *EMBO Mol Med* (2014), et al. In this manuscript, our *in vivo* studies clearly suggest that human Treg cells can induce T cell senescence, which can be prevented *via* the inhibition of the DNA damage response and/or STAT signaling in NSG mouse models.

I do understand and agree with this reviewer's comment about the potential artificial environment in NSG mice. However, **there is no other alternative way** to directly study human cells in animal models. In addition, humanized mouse models have now been well recognized and utilized in the research field. Actually, our results and studies using humanized mouse models with human cells are more clinically relevant.

2) The authors assume that normal human T cells can be categorized into senescent, exhausted and anergic T cell subtypes based on their distinct phenotypic and in part functional attributes. The authors go as far as to state that nTreg inducing senescence via ATM-associated DNA damage response represent a distinct lineage of T cells. There is no evidence for this, nor is there convincing evidence for the existence under normal physiological conditions of the three T-cell categories, as a considerable degree of overlap is evident.

Response: I understand this reviewer's comment. The purpose of our studies is to characterize the phenotypes and function of senescent T cells induced by human Treg cells, and then further to dissect potential molecular mechanisms responsible for the senescence induction/generation.

The exhausted and anergic T cell subtypes have been brought up by the many other research groups in the field. I do not think that it is necessary to provide convincing evidence for the existence under normal physiological conditions of the three T-cell categories, as a considerable degree of overlap, which might be totally depended on the local microenvironment within the hosts. In addition, this is also out of the scope of our current study. In fact, recent studies suggest that senescence also occurs in human T cells, causing age-associated dysregulation of the immune system during the normal aging process^{8,9}. Furthermore, accumulation of senescent CD8⁺ T cells has also been found in younger patients with chronic viral infections, as well as patients with certain types of cancers¹⁰⁻¹⁷.

Collectively, our current studies identify novel molecular processes responsible for human Treg suppression and provide an emerging concept that targeting Treg-induced T cell senescence is a new checkpoint for immunotherapy against cancer and other Treg-associated diseases as well. We will continue to test this novel concept with different therapeutic models in the future.

3) The authors are working with nTreg isolated by sorting of CD4+CD25hi T cells. This is a highly biased approach to the isolation of Treg. What is the complete phenotype of the isolated nTreg? Are highly activated CD4+ cells selected as well as nTreg?? These isolated nTreg are then placed in long-term co-cultures with Teff and plenty of possibilities for selective interactions, given the well-known plasticity of human T cells. So the cellular origin and unique features of the nTreg inducing senescence in Teff are open to criticism.

Response: I understand this reviewer's concern, but I respectfully disagree with this reviewer's comment. I must clarify the origin and functional characteristics of nTreg cells utilized in this manuscript.

It is well recognized that CD4⁺CD25^{hi} T cells from both mice and humans belong to Treg cells. This strategy has been confirmed and utilized by the researchers in the field for the past 2 decades. The nTreg cells utilized in the performed experiments in this manuscript were purified by FACS sorting based on CD25 high expression (around 1%). The purified nTreg cells were further confirmed by phenotypic analysis (FoxP3 expression and demethylation status) and suppressive activity on other T cell proliferation, before we used it for our studies.

In addition, the activated CD4⁺ effector T cells are not Treg cells, although they can express CD25 and FoxP3 (usually at the medium expression levels). In addition, the FoxP3 in activated T cells is not demethylated. Furthermore, activated T cells themselves cannot become a suppressive Treg cells in a short time culture (3-5 days). I agree with this reviewer's opinion about the plasticity of human T cells, but it happens to be in a certain local environment or cytokine-driven condition. In fact, we have already fully characterized how the effector T cells become senescent T cells co-cultured with Treg cells in our recently published papers (*Blood* 2012 and *J Immunol* 2013). Furthermore, all our studies have included the other cell controls with the same conditions.

To address this reviewer's concern, we provide the data for how to purify and characterize nTreg cells utilized in this manuscript. We purified the naturally occurring CD4⁺ Treg cells with highest expression of CD25 from healthy donors (top 1% of positive cells). We further confirmed that these CD4⁺CD25^{hi} populations were almost all FoxP3⁺ T cells (**Attached Figure 2A**). FoxP3 is the most specific molecular marker for Tregs, but it is also transiently expressed in activated conventional T cells. Recent studies have shown that human FoxP3 contains several highly conserved demethylation regions that are exclusive for Treg cells. We thus compared the FoxP3 methylation levels in CD4⁺CD25^{hi}FoxP3⁺, CD4⁺CD25⁻, and anti-CD3-activated naïve CD4⁺ T cells. As expected, the TSDR within the FoxP3 locus of CD4⁺CD25^{hi}FoxP3⁺ T cells was almost completely demethylated compared with that of CD4⁺CD25⁻ T cells and anti-CD3-activated naïve T cells (**Attached Figure 2B**). We then investigated the suppressive capacity of CD4⁺CD25^{hi}FoxP3⁺ Treg cells using functional proliferation assays. As shown in **Attached Figure 2C**, CD4⁺CD25^{hi}FoxP3⁺ Treg cells strongly inhibited the proliferation of naïve CD4⁺ T cells in the presence of anti-CD3 antibody using [³H]-thymidine incorporation assays. We also further confirmed the suppression mediated by human CD4⁺CD25^{hi}FoxP3⁺ naturally occurring Treg cells using CFSE dilution assays (**Attached Figure 2D**). **These results suggest that the**

human CD4⁺CD25^{hi}FoxP3⁺ T cell populations are Treg cells with potent suppressive activity.

Attached Figure 2. Purification of functional CD4⁺CD25^{hi}FoxP3⁺ Treg cells from PBMCs.

(A) Isolation of CD4⁺CD25^{hi}FoxP3⁺ Treg and CD4⁺CD25⁻ effector T cells from PBMCs of healthy donors by FACS sorting. FoxP3 expression in the isolated T cells was further confirmed by FACS analyses. (B) Relative FoxP3 methylation levels of different T cells were determined by real-time quantitative PCR with methylation-specific primers, and normalized to β-actin expression and compared with the expression level of methylated FoxP3 in

CD4⁺CD25⁻ T cells. CD4⁺CD25⁻ and anti-CD3-activated CD4⁺ T cells were included as controls. All experiments were performed in triplicate. (C) and (D) Suppression of naïve T cell proliferation by CD4⁺CD25^{hi}Foxp3⁺ Tregs. CD4⁺CD25⁻ effector T cells served as a negative control displaying no suppressive activity. Naïve CD4⁺ T cells were co-cultured with Treg or control T cells at a ratio of 10:1. The proliferation of naïve CD4⁺ T cells in the presence of anti-CD3 antibody was determined by [³H]-thymidine incorporation assays (C) or CFSE dilution assays (D).

4) The authors are working with nTreg derived from normal PBMC. Yet, the text is peppered with comments to tumor-reactive T cells and tumor-associated Treg. Based on the presented data no comments relevant to T cells in cancer or those in human tumors can be made. Similarly, references to breast cancer-derived γ/δ Treg made in the text bear no relationship to the presented data. Treg subsets in cancer patients or those in patients with chronic infections are not comparable to nTreg populating PBMC of normal donors.

Response: We have been working on different types of human Treg cells, including nTreg cells in the tumor suppressive microenvironment. The current studies are based on our previous work and further identify the molecular mechanisms mediated by human Treg cells [*Science* (2005), *Immunity* (2004 & 2007), *Blood* (2012), *PNAS* (2008), *EMBO Molecular Medicine* (2014), *Cancer Research* (2005, 2008 & 2013), and *Journal Immunol* (2005, 2010, 2012 & 2013)]. Based on our studies, we have obtained the following insights: 1) Different types of Treg cells may be through (use) different molecules/pathways to induce immune suppression in targeted T cells. However, induction of targeted cell senescence is the common step for their suppression. 2) We found the suppressive cytokines IL-10, IL-35 and TGF- β , as well as IDO are not involved in the induction of human T cell senescence induced by human Treg cells, although these might be important for mouse Treg-mediated immune suppression. 3) Our previous studies have identified that tumor-derived cAMP is responsible for tumor-induced immune suppression and senescence in T cells. However, we found that cAMP is not the cause for human Treg suppression and responder T cell senescence. We have clarified the origins and sources of tumor-derived Treg cells in the revised manuscript.

In addition, I respectively disagree with this reviewer's opinion that "Treg subsets in cancer patients or those in patients with chronic infections are not comparable to nTreg populating PBMC of normal donors". There is no evidence suggest that human nTreg cells and tumor-derived Treg cells utilize different mechanisms to perform their suppressive function so far, especially the differences in the fate and function of suppressed responder T cells. Furthermore, majority of our current knowledge and information about Treg cell suppressive mechanism studies are derived from mouse models. Very limited information is known about human Treg cell suppression, especially tumor-derived Treg cells from cancer patients. Our group has been focusing on these research challenging issues during the past years. Our recent studies have identify that induction of responder T cell senescence is a novel suppressive mechanism utilized by both human nTreg cells and breast cancer-derived gamma/delta Treg cells (*Blood* 2012 and *J Immunol* 2013).

5) Overall, while the possibility that human nTreg generated in vitro are able to mediate suppression via ATM-associated DNA damage response is not questioned, the biological significance of this type of regulation operating in vivo remains unclear. The attempt to treat nTreg that use this mechanism of suppression in vitro as a unique subset, or a distinct lineage, of

Treg and to compare or contrast the senescent-inducing nTreg to those inducing exhaustion or anergy is not supported by the presented data. The categories of Treg and mechanisms of suppression these cells utilize are context dependent and likely vary broadly between health and disease.

Response: We understand this reviewer’s concern. Our recent studies have already identify that induction of responder T cell senescence is a novel suppressive mechanism utilized by both human nTreg cells and breast cancer-derived gamma/delta Treg cells (*Blood* 2012 and *J Immunol* 2013). In this manuscript, we have further identified the novel molecular mechanism that human Treg cells initiate the ATM-associated DNA damage response in responder T cells during their cross-talk, resulting in responder T cell senescence and functional changes. We then identified that MAPK ERK1/2 and p38 signaling functionally cooperate with the transcription factors STAT1/STAT3 to control the molecular process of responder T cell senescence induced by human Treg cells. Importantly, our *in vivo* studies suggest that human Treg-induced T cell senescence can be prevented *via* the inhibition of the DNA damage response and/or STAT signaling in NSG mouse models.

We acknowledge that it is difficult to perform the experiments *in vivo* using human T cells and tumor cells to test the hypothesis and clinical relevance in mouse models. **However, significant accumulation of senescent CD8⁺ T cells has been found in tumor-infiltrating T cells (TILs)** from various types of cancer patients, including lung¹⁰, colorectal¹⁸, endometrial¹⁹, ovarian²⁰, lymphoma²¹ and breast cancers^{17,22}, as well as in metastatic satellite lymph nodes and peripheral blood of cancer patients^{10,23}. Our recent data from tumor-infiltrating T cells obtained from both human cancer patients and tumor-bearing mice clearly demonstrated that high percentages of senescent T cells exist in the tumor suppressive microenvironment, further suggesting that targeting T cell senescence could be an effective and important strategy for tumor immunotherapy.

(1) Prevalence of senescent T cell populations in tumor suppressive microenvironments of cancer patients. We generated TIL lines from fresh tumor tissues obtained from breast cancer, head and neck cancer, and melanoma patients. In parallel, tumor-matched breast tissue-infiltrating T cell lines from normal breast tissues were also generated. We observed markedly elevated SA-β-Gal positive T cells existing in TILs derived from melanoma, head and neck, and breast cancer patients (over 25%) (**Attached Figure 3A**). In contrast, SA-β-Gal positive cell populations among the normal breast tissue-infiltrating T cells constituted less than 10%. We determined whether TILs also down-regulated expression of the co-stimulatory molecule CD28. We found that both normal breast and melanoid tissue-infiltrating T lymphocytes expressed high levels of CD28. However, TILs derived from the matched cancer patients significantly down-regulated CD28 expression, further confirming the existence of elevated senescent T cell populations in the tumor microenvironments (**Attached Figure 3B**).

Attached Figure 3. Accumulation of senescent T cells in the tumor suppressive microenvironment.

(A) and (B) show that increased SA-β-Gal positive T cell populations (A) and decreased CD28 expression (B) existed in

TILs, compared with those in normal tissue-derived lymphocytes. CD3⁺ T cells were purified from the cultured tissue-infiltrating T cells from freshly digested tumor or normal tissues by microbeads and SA- β -gal staining or CD28 expression was determined. BNT: Normal breast tissue-derived T cells; BTIL, MTIL, and HNTIL: TIL obtained from breast cancer, melanoma and head and neck cancers, respectively.

(2) Existence of elevated senescent T cell populations in tumor-bearing mice. We further investigated whether senescent T cells also exist *in vivo* in the tumor-bearing mice. We utilized mouse melanoma B16F10 and E0771 breast cancer tumor models to test the hypothesis. Our results suggest that significantly increased SA- β -Gal⁺ cell populations were induced in both CD4⁺ and CD8⁺ T cells obtained from different organs and tumors in both tumor bearing mouse models (**Attached Figure 4**).

Attached Figure 4. Elevated senescent T cell populations in tumor-bearing mice. (A) Increased SA- β -Gal⁺ cell populations were induced in CD4⁺ and CD8⁺ T cells purified from different organs and tumor tissues in melanoma-bearing mice. B16F0 (2×10^5 cells/mouse) were implanted into C57BL/6 mice. When primary tumors reached 10-15 mm in diameters, the tumor-bearing mice and tumor free-littermate controls were sacrificed. Blood, lymph nodes (LN), spleens (SP) and tumor tissues were harvested and CD4⁺ and CD8⁺ T cells were purified for SA- β -Gal staining. (B) Increased SA- β -Gal⁺ cell populations were induced in CD4⁺ and CD8⁺ T cells from different organs in breast cancer-bearing mice. E0771 (2×10^5 cells/mouse) were implanted into mammary gland fat pads of female C57BL/6 mice. When primary tumors reached 10-15 mm in diameters, the tumor-bearing mice and tumor free-littermate controls were sacrificed. Cell purification and staining were the same as in (A). ** $p < 0.01$, compared with the populations from WT mice; and # $p < 0.01$, compared with the populations from T cells in WT mice.

6. The manuscript contains numerous grammatical errors which should be corrected.

Response: We have carefully checked and corrected in this revised manuscript.

References

1. Ye, J., *et al.* Human regulatory T cells induce T-lymphocyte senescence. *Blood* **120**, 2021-2031 (2012).
2. Ye, J., *et al.* Tumor-Derived gammadelta Regulatory T Cells Suppress Innate and Adaptive Immunity through the Induction of Immunosenescence. *J Immunol* **190**, 2403-2414 (2013).
3. Hawley, S.A., *et al.* Use of cells expressing gamma subunit variants to identify diverse mechanisms of AMPK activation. *Cell metabolism* **11**, 554-565 (2010).
4. Fu, X., Wan, S., Lyu, Y.L., Liu, L.F. & Qi, H. Etoposide induces ATM-dependent mitochondrial biogenesis through AMPK activation. *PLoS one* **3**, e2009 (2008).
5. Lanna, A., Henson, S.M., Escors, D. & Akbar, A.N. The kinase p38 activated by the metabolic regulator AMPK and scaffold TAB1 drives the senescence of human T cells. *Nat Immunol* **15**, 965-972 (2014).
6. Chang, C.H., *et al.* Metabolic Competition in the Tumor Microenvironment Is a Driver of Cancer Progression. *Cell* **162**, 1229-1241 (2015).
7. Sukumar, M., Roychoudhuri, R. & Restifo, N.P. Nutrient Competition: A New Axis of Tumor Immunosuppression. *Cell* **162**, 1206-1208 (2015).
8. Effros, R.B., Dagarag, M., Spaulding, C. & Man, J. The role of CD8+ T-cell replicative senescence in human aging. *Immunol Rev* **205**, 147-157 (2005).
9. Weng, N.P., Akbar, A.N. & Goronzy, J. CD28(-) T cells: their role in the age-associated decline of immune function. *Trends Immunol* **30**, 306-312 (2009).
10. Meloni, F., *et al.* Foxp3 expressing CD4+ CD25+ and CD8+CD28- T regulatory cells in the peripheral blood of patients with lung cancer and pleural mesothelioma. *Hum Immunol* **67**, 1-12 (2006).
11. Tsukishiro, T., Donnenberg, A.D. & Whiteside, T.L. Rapid turnover of the CD8(+)/CD28(-) T-cell subset of effector cells in the circulation of patients with head and neck cancer. *Cancer Immunol Immunother* **52**, 599-607 (2003).
12. Wolfram, R.M., *et al.* Defective antigen presentation resulting from impaired expression of costimulatory molecules in breast cancer. *International journal of cancer* **88**, 239-244 (2000).
13. Chen, W.H., *et al.* Vaccination in the elderly: an immunological perspective. *Trends Immunol* **30**, 351-359 (2009).
14. Appay, V., *et al.* HIV-specific CD8(+) T cells produce antiviral cytokines but are impaired in cytolytic function. *J Exp Med* **192**, 63-75 (2000).
15. Montes, C.L., *et al.* Tumor-induced senescent T cells with suppressor function: a potential form of tumor immune evasion. *Cancer Res* **68**, 870-879 (2008).
16. Melichar, B., Touskova, M., Dvorak, J., Jandik, P. & Kopecky, O. The peripheral blood leukocyte phenotype in patients with breast cancer: effect of doxorubicin/paclitaxel combination chemotherapy. *Immunopharmacol Immunotoxicol* **23**, 163-173 (2001).
17. Schule, J.M., Bergkvist, L., Hakansson, L., Gustafsson, B. & Hakansson, A. CD28 expression in sentinel node biopsies from breast cancer patients in comparison with CD3-zeta chain expression. *Journal of translational medicine* **2**, 45 (2004).
18. Ye, S.W., *et al.* Ex-vivo analysis of CD8+ T cells infiltrating colorectal tumors identifies a major effector-memory subset with low perforin content. *Journal of clinical immunology* **26**, 447-456 (2006).

19. Chang, W.C., *et al.* Clinical significance of regulatory T cells and CD8+ effector populations in patients with human endometrial carcinoma. *Cancer* **116**, 5777-5788 (2010).
20. Webb, J.R., *et al.* Profound elevation of CD8+ T cells expressing the intraepithelial lymphocyte marker CD103 (alphaE/beta7 Integrin) in high-grade serous ovarian cancer. *Gynecologic oncology* **118**, 228-236 (2010).
21. Urbaniak-Kujda, D., *et al.* Increased percentage of CD8+CD28- suppressor lymphocytes in peripheral blood and skin infiltrates correlates with advanced disease in patients with cutaneous T-cell lymphomas. *Postepy higieny i medycyny doswiadczonej* **63**, 355-359 (2009).
22. Gruber, I.V., *et al.* Down-regulation of CD28, TCR-zeta (zeta) and up-regulation of FAS in peripheral cytotoxic T-cells of primary breast cancer patients. *Anticancer research* **28**, 779-784 (2008).
23. Filaci, G., *et al.* CD8+ CD28- T regulatory lymphocytes inhibiting T cell proliferative and cytotoxic functions infiltrate human cancers. *Journal of immunology* **179**, 4323-4334 (2007).

REVIEWERS' COMMENTS:

Reviewer #1 (Remarks to the Author):

I appreciate the effort of the authors to perform new experiments as requested and to clarify our minor points. Regarding the answer to point #9 (and Referee 2 point #1) an attached Figure with additional data is presented. It is my understanding that this figure won't be present in the final manuscript.

However, this new set of data are critical to elucidate the molecular link between glucose intake and senescence. My suggestion is that this data have to be included in this paper in order to be suitable for publication in Nature Communications

Reviewer #2 (Remarks to the Author):

The authors have added a significant amount of nice data to the manuscript that significantly strengthens the manuscript. They now convincingly show that:

- (1) low glucose levels induce a DNA damage response in responder T cells
- (2) Tregs have high levels of glucose uptake and can deplete glucose from the culture media over time.
- (3) In this in vitro system, Treg depletion of glucose from the culture is promoting senescence in the responder T cells.

However, all this data needs to be interpreted on the basis that the authors are using a closed tissue culture system over a long period of time (3 days). This does not accurately recreate the in vivo situation where nutrients are continuously supplied from the circulation.

The authors state that "our current study provides the first evidence that human Treg cells directly suppress responder T cell proliferation and initiate the ATM-associated DNA damage response triggered by glucose competition, resulting in the senescence and functional changes in T cells." Due to the limitations of the study discussed above the authors are overstating their results. I agree that glucose deprivation is likely to be a mechanism to induce T cell senescence via a DNA damage response in vivo, but the low glucose levels in tumours are due to the high glucose consumption rates of the abundant tumour cells rather than the few Tregs. I appreciate that there are high numbers of nTreg and gdTregs in tumour tissue but these cells are still significantly outnumbered by the tumour cells.

The authors should acknowledge and discuss the limitations of the approach in the manuscript and adjust their conclusions to more accurately portray the contribution of nTregs or gdTregs to low glucose-induced senescence in Cd4 T cells.

Reviewer #3 (Remarks to the Author):

The authors have provided an insightful response to the critique. A number of additional experiments were performed adding strength to the authors' favorite hypothesis for the mechanism explaining Treg suppressive functions. The authors very convincingly argued for this mechanism as one that applies to nTreg as well as iTreg in diseased tissues. I have to respect their arguments and concur with the importance of the metabolic-based mechanism they demonstrate. The revised manuscript deserves to be published.

RE: MS# NCOMMS-17-01271B-Z, “Regulatory T cells trigger a DNA damage response and senescence in responder T cells molecularly distinct from anergic and exhausted T cells”

Dear Reviewers:

We would like to thank you very much for your efforts in reviewing above our manuscript (NCOMMS-17-01271B-Z). We appreciate your enthusiastic and positive comments, which are very critical and helpful for improving our work. To address your concerns and insightful suggestions, we have included new Supplementary Figure 4 and provided a discussion about the limitations of our studies in this revised manuscript. These new data and changes have been incorporated into the main text, figure legends and figures in this revised manuscript. In addition, a point-by-point reply to the reviewers’ concerns is included below.

Reviewer #1

I appreciate the effort of the authors to perform new experiments as requested and to clarify our minor points. Regarding the answer to point #9 (and Referee 2 point #1) an attached Figure with additional data is presented. It is my understanding that this figure won’t be present in the final manuscript.

However, this new set of data is critical to elucidate the molecular link between glucose intake and senescence. My suggestion is that this data have to be included in this paper in order to be suitable for publication in Nature Communications.

Response: We appreciate the helpful suggestion from this reviewer for improving our manuscript. We have included our newly generated data into the **Supplementary Figure 4** and described in **Page 9, lines 12-22** in this revised manuscript

Reviewer #2

The authors have added a significant amount of nice data to the manuscript that significantly strengthens the manuscript. They now convincingly show that: (1) low glucose levels induce a DNA damage response in responder T cells. (2) Tregs have high levels of glucose uptake and can deplete glucose from the culture media over time. (3) In this in vitro system, Treg depletion of glucose from the culture is promoting senescence in the responder T cells.

However, all this data needs to be interpreted on the basis that the authors are using a closed tissue culture system over a long period of time (3 days). This does not accurately recreate the in vivo situation where nutrients are continuously supplied from the circulation.

The authors state that “our current study provides the first evidence that human Treg cells directly suppress responder T cell proliferation and initiate the ATM-associated DNA damage response triggered by glucose competition, resulting in the senescence and functional changes in T cells.” Due to the limitations of the study discussed above the authors are overstating their results. I agree that glucose deprivation is likely to be a mechanism to induce T cell senescence via a DNA damage response in vivo, but the low glucose levels in tumours are due to the high glucose consumption rates of the abundant tumour cells rather than the few Tregs. I appreciate that there are high numbers of nTreg and gdTregs in tumour tissue but these cells are still significantly outnumbered by the tumour cells.

The authors should acknowledge and discuss the limitations of the approach in the manuscript and adjust their conclusions to more accurately portray the contribution of nTregs or gdTregs to low glucose-induced senescence in CD4 T cells.

Response: We thank this reviewer for his insightful suggestions. We fully agree with his comments and acknowledge that our identified mechanism responsible for human Treg suppression in this manuscript is mainly based on the in vitro studies in the co-culture system. As suggested by this reviewer, we have included a discussion about the limitations and future studies using different tumor models in the Discussion section in this revised manuscript (**Page 19, lines 8-15**).

Reviewer #3

The authors have provided an insightful response to the critique. A number of additional experiments were performed adding strength to the authors' favorite hypothesis for the mechanism explaining Treg suppressive functions. The authors very convincingly argued for this mechanism as one that applies to nTreg as well as iTreg in diseased tissues. I have to respect their arguments and concur with the importance of the metabolic-based mechanism they demonstrate. The revised manuscript deserves to be published.

Response: We would like to thank this reviewer again for his favored review of our manuscript.